# Cerebellar control of targeted tongue movements

Lorenzo Bina[1], Camilla Ciapponi[1], Si-yang Yu[1], Xiang Wang[1], Laurens W. J. Bosman[1] (ID) and Chris I. De Zeeuw[1,2] (ID)

[1] *Department of Neuroscience, Erasmus MC, Rotterdam, The Netherlands*
[2] *Netherlands Institute for Neuroscience, Royal Academy of Sciences, Amsterdam, The Netherlands*

Handling Editors: Harold Schultz & Jing-Ning Zhu

The peer review history is available in the Supporting Information section of this article (https://doi.org/10.1113/JP287732#support-information-section).

**Abstract figure legend** Cerebellar Purkinje cells produce two types of action potentials: complex spikes (green asterisk) and simple spike (purple stripes). We recorded Purkinje cell activity in mice that are engaged in licking trials. We observed that complex spikes related only weakly to individual, unperturbed licks, but reacted strongly to sudden changes in the target location for licking. This peak in complex spike activity was followed by an increase in simple spikes that induced a bending of the tongue. Thus, the combination of complex spike and simple spike responses enable correct targeting of the tongue, in particular when the circumstances change.

This manuscript was first published as a preprint. Bina L, Ciapponi C, Yu S, Wang X, Bosman L, De Zeeuw CI. 2024. Cerebellar control of targeted tongue movements. bioRxiv. https://doi.org/10.1101/2024.09.26.615128

**Abstract** The cerebellum is critical for coordinating movements related to eating, drinking and swallowing, all of which require proper control of the tongue. Cerebellar Purkinje cells can encode tongue movements, but it is unclear how their simple spikes and complex spikes induce changes in the shape of the tongue that contribute to goal-directed movements. To study these relations, we recorded and stimulated Purkinje cells in the vermis and hemispheres of mice during spontaneous licking from a stationary or moving water spout. We found that Purkinje cells can encode rhythmic licking with both their simple spikes and complex spikes. Increased simple spike firing during tongue protrusion induces ipsiversive bending of the tongue. Unexpected changes in the target location trigger complex spikes that alter simple spike firing during subsequent licks, adjusting the tongue trajectory. Furthermore, we observed increased complex spike firing during behavioural state changes at both the start and the end of licking bouts. Using machine learning, we confirmed that alterations in Purkinje cell activity accompany licking, with different Purkinje cells often exerting heterogeneous encoding schemes. Our data highlight that directional movement control is paramount in cerebellar function and that modulation of the complex spikes and that of the simple spikes are complementary during acquisition and execution of sensorimotor coordination. These results bring us closer to understanding the clinical implications of cerebellar disorders during eating, drinking and swallowing.

(Received 25 September 2024; accepted after revision 10 January 2025; first published online 26 January 2025)

**Corresponding authors** L. W. J. Bosman and C. I. De Zeeuw: Department of Neuroscience, Erasmus MC, Rotterdam, The Netherlands. Email: l.bosman@erasmusmc.nl and c.dezeeuw@erasmusmc.nl

**Key points**

- When drinking, mice make rhythmic tongue movements directed towards the water source.
- Cerebellar Purkinje cells can fire rhythmically in tune with the tongue movements.
- Purkinje cells encode changes in the position of the water source with complex spikes.
- Purkinje cell simple spike firing affects the direction of tongue movements.
- Purkinje cells that report changes in the position of the target can also adjust movements in the right direction.

## Introduction

Cortical and subcortical structures, like the cerebellum and basal ganglia, are essential for controlling complex motor behaviours that necessitate the coordinated action and precise sequencing of multiple muscle groups. This holds not only for skilled movements of the whole body (Kim et al., 2017; Sauerbrei et al., 2015; Ting et al., 2015; Vinueza Veloz et al., 2015), but also for those of individual limbs (Diedrichsen et al., 2005; Park et al., 2022) or the tongue (Cullins et al., 2019; Leopold & Kagel, 1996; Travers et al., 1997). Indeed, various neurological conditions can compromise complex motor behaviours, and tongue movements are no exception: dystonia, dyskinesia and/or tremor of the tongue, as well as impaired oral sensation, can impair such crucial actions as mastication, drinking, swallowing, breathing and speaking (Ghadery et al., 2022; Laurence-Chasen et al., 2022; Leopold & Kagel, 1996). Eventually, improper control of the tongue can lead to the life-threatening

**Lorenzo Bina** is finishing his PhD at the Neuroscience department of Erasmus Medical Center, Rotterdam, the Netherlands, where he worked as researcher for the past 8 years, since 2016. Originally from a village on the Lake Maggiore, in the north of Italy, he obtained his bachelor's degree in Biological Sciences and master's degree in Neurobiology at the University of Pavia, Italy, where he started training in electrophysiology. Today he makes use of extracellular recordings, optogenetics or calcium imaging in awake mice, to investigate the role of cerebellar Purkinje cells in learning and execution of tasks requiring cognitive and motor skills.

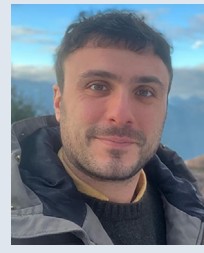

condition of aspiration pneumonia (Krohn et al., 2023; Takizawa et al., 2016).

Accordingly, patients suffering from diseases of the cerebellum, such as hereditary cerebellar ataxia caused by malfunction of cerebellar Purkinje cells, can show symptoms like dysarthria, dysphagia or choking due to reduced motor control of the tongue (Giardina et al., 2020; Ikeda et al., 2012; Keage et al., 2017; Markovic et al., 2016; Rezende Filho et al., 2019; Ushe & Perlmutter, 2012; Woo et al., 2019). In line with the latter, functional imaging in human subjects has revealed specific activation patterns in the cerebellar cortex and nuclei during both voluntary and subconscious tongue movements (Corfield et al., 1999; Dimitrova et al., 2006; Groenendijk et al., 2020; Ogura et al., 2012; Sörös et al., 2020), while electrophysiological recordings in Purkinje cells and other cerebellar neurons during licking have shown close correlations between cerebellar activity and tongue movements (Bina et al., 2021; Bryant et al., 2010; Gaffield et al., 2022; Gao et al., 2018; Lackey et al., 2024; Lu et al., 2013; Welsh et al., 1995). However, despite the experimental and clinical evidence for a role of the cerebellum in controlling tongue movements, our understanding of the mechanisms by which neural activity of cerebellar neurons can affect tongue movements remains limited.

The primary driver of tongue movements is a central pattern generator consisting of premotor neurons in the intermediate reticular formation of the brainstem (Brozek et al., 1996; Dempsey et al., 2021; Kleinfeld et al., 2023; Travers et al., 1997). These reticular neurons directly activate motor neurons in the hypoglossal nucleus, which control all four intrinsic and three out of the four extrinsic tongue muscles, while they may indirectly also affect the vagal nerve that drives the fourth extrinsic muscle (Altschuler et al., 1994; Guo et al., 2020; Wiesenfeld et al., 1977). As a consequence, a lack of the cerebellar output that influences the activity of the reticular neurons reduces the efficiency of tongue movements (Bryant et al., 2010), while it does not preclude making tongue movements *per se* (Timmann et al., 2003). Given the multitude of muscles required for precise tongue movements, we hypothesize that the cerebellum is required for the coordination of these muscles, similar to its function of inter-limb coordination to allow walking over a horizontal ladder (Jaarsma et al., 2024; Vinueza Veloz et al., 2015).

If the cerebellum is indeed able to coordinate the activation of the tongue muscles so that tongue movements can be adapted to changes in the behavioural needs, then Purkinje cells should encode unperturbed tongue movements, react to environmental changes and translate these into altered movements. Using statistical analysis and a machine learning model, we demonstrate that Purkinje cells indeed encode spontaneous licking. In particular, complex spikes occur during behavioural state changes and thus at the start and end of licking bouts, while simple spikes are associated with individual licks. In addition, complex spikes report fast changes in the position of the water spout, and this is correlated to alterations in the simple spike firing, causing a rapid adaptation of the tongue trajectory. Finally, using optogenetic stimulation of Purkinje cells, we could demonstrate the causal impact of simple spike firing on the tongue trajectory. Thus, cerebellar Purkinje cells meet the criteria for mediating adaptation of tongue movements to changes in environmental demands.

## Methods

### Ethical approval

All experimental procedures involving animals were evaluated before the start of this study by an independent animal ethical committee (DEC-Consult, Soest, The Netherlands) and subsequently approved by the national authority (Centrale Commissie Dierproeven, The Hague, The Netherlands; project license number AVD1010020197846). Compliance with the project license of each experiment was confirmed by the animal welfare body of Erasmus MC, and all experiments were performed following the relevant institutional regulations of Erasmus MC, Dutch legislation on animal experimentation and EU directive 2010/63/EU. Throughout the study, care was taken to minimize pain and suffering. The authors understand and comply with the animal ethical principles of *The Journal of Physiology*.

### Mice

All experiments that did not involve optogenetic stimulation were performed with C57BL/6J mice (Charles River, 's Hertogenbosch, The Netherlands). For experiments with optogenetic stimulation, Tg(Pcp2-cre)2Mpin; Gt(ROSA)26Sor[tm27.1(CAG-OP4*H134R/tdTomato)Hze] mice on a C57BL/6J background were used (Witter et al., 2013), and these mice were bred in the animal facility of the Erasmus MC with regular back-crossings with wild-type C57BL/6J mice. All mice were adults between 12 and 35 weeks of age and specific pathogen free. The animals were group-housed until surgery, after which they were single-housed to avoid inflicting wounds by cage mates. Mice were kept in a vivarium with controlled temperature and humidity and a 12/12 h light–dark cycle.

The mice had *ad libitum* access to standard food and water at all times, except when they were trained to lick from the water spout in the recording set-up. During training, mice had access to water only during the daily sessions, while food remained available *ad libitum*. If a mouse did not drink sufficiently in the set-up, it received the daily dosage of water (1 ml/20 g body weight) about half an hour after the training session. Daily weighing

was used to monitor putative weight loss, which was limited at 20% of the original body weight according to the restrictions in our project license. Typically, however, mice did not lose that much weight: after a first drop of about 10–15% in the first 5 days, the body weight typically became stable or even increased once mice got used to drink *ad libitum* from the set-up and their metabolism adapted.

A total of 39 mice were used for electrophysiological recordings, nine of which were trained in the motor adaptation task, and eight mice were used for optogenetic stimulation. We used approximately equal numbers of male and female mice. Given the simplicity of the behavioural task, all mice reached the final stage of the experiments. Only in rare cases when the electrophysiological recordings were not successful, were mice excluded from the animal count. At the end of the final experimental session, mice were euthanized by cervical dislocation under isoflurane anaesthesia.

## Surgical procedures

Surgeries were performed under isoflurane anaesthesia (induction: 4% v/v; maintenance: 2% v/v in $O_2$). Before the start of the surgery, the depth of anaesthesia was verified by the absence of a reaction to an ear pinch. When the depth of anaesthesia was sufficient, the mouse was put on a heating pad to keep its body temperature stable at 37°C, which was verified with a rectal thermometer. The eyes were protected with eye ointment (Duratears, Alcon, Fort Worth, TX, USA). To prevent dehydration, each mouse received 1 ml of saline s.c. injection before the surgery commenced.

Mice received a magnetic pedestal for head fixation, attached to the skull above bregma using Optibond adhesive (Kerr Corporation, Orange, CA, USA). To place this pedestal, the surgical spot was shaved and lidocaine (4 mg/kg s.c.; Braun, Melsungen, Germany) was injected. After 1 min, a small incision was made in the skin so that the periosteum became visible. A few extra drops of 2% lidocaine were applied topically, after which the periosteum was removed and the skull was cleaned and dried.

Subsequently and while still under anaesthesia, a craniotomy was performed to expose the recording area (the lateral part of vermal lobules VI an VII and the adjacent hemispheric lobules crus 1 and crus 2 on the right side of the mouse). Four mice received a larger and more central craniotomy in order to expose both right and left vermis for optogenetic stimulation. The preparation was identical as for the pedestal placement, including shaving, lidocaine injection, and a small incision in the skin. With a dental drill, an opening in the skull was made, while the dura mater was preserved. The craniotomy was cleaned, and a small chamber was built with dental cement surrounding the craniotomy to contain fluids during the

later recording session. At the end of the surgery, the craniotomy was covered with Kwik-Cast (World Precision Instruments, Sarasota, FL, USA). Postsurgical pain was treated with carprofen (5 mg/kg s.c.; Pfizer, New York, NY, USA), buprenorphine (50 µg/kg s.c.; Indivior, Richmond, VA, USA) and bupivacaine (1 mg/kg s.c.; Actavis, Parsipanny-Troy Hills, NJ, USA).

At the end of surgery, the mice were allowed to recover under a warming lamp for at least 30 min. The warming lamp covered only a part of the cage, so that the mice could chose to be under the lamp or not. When a mouse was behaving normally again, it was brought back to the stable.

## Habituation

After recovery from surgery, mice were handled daily by the experimenter to reduce stress. When mice were relaxed when handling; they were head-fixed for ∼15 min a day during which water was available from the lick-port positioned in front of the mouse.

## Lick detection

During training and experimental sessions, mice had access to a water spout in front of their mouth. At this water spout, water was freely available. Licking was detected using an optical sensor placed 2 mm before the water spout. In a subset of experiments, tongue trajectories were recorded with an overhead video camera (frame rate 100 Hz). The coordinates of the tip of the tongue were extracted using DeepLabCut software for pose estimation (Mathis et al., 2018). We used the location of maximal protrusion for further analysis. From the video analysis it became clear that the moment of maximal protrusion was on average achieved 16 (2) ms after detection by the optical sensor. Licking bouts were defined as sequences of licks with intervals <500 ms.

## Moving lick-port

To test the ability of mice to target their tongue movements, we trained nine mice to lick from the same lick-port as used in the other experiments, but for these mice, we moved the lick-port 3 mm to the right at unpredictable moments. At the start of each session, the lick-port was in front of the mouse. We used a closed-loop control circuit to detect licking. Once a lick was detected, there was a 50% chance to trigger a rightward movement, starting 40 ms after lick detection and with a travel time of 50 ms. For the movement, we used a piezo actuator (Physik Instrumente, Karlsruhe, Germany). In this way, we moved the lick-port during retraction of the tongue. The lick-port was moved back to the central position after 750 ms and stayed there for at least 750 ms before it was available for

another movement as triggered by the closed-loop control circuit. After a week of training, mice were able to adjust the direction of their tongue movements to follow the lick-port.

### Electrophysiological recordings

Prior to the recordings, the mice were anaesthetized with isoflurane (induction: 4% v/v; maintenance: 2% v/v in $O_2$). Before continuing, the depth of anaesthesia was verified by the absence of a reaction to an ear pinch. If the depth of anaesthesia was sufficient, the dura mater was removed using very fine forceps. Afterwards, the mouse was head-fixed in the set-up using the previously placed magnetic pedestal. Next, we placed the recording electrodes on the surface of the cerebellum, and let the mouse recover from anaesthesia in the set-up. To minimize the impact of anaesthesia, we did not start recording within 60 min after recovery from anaesthesia. Once well awake, the attention of the mice was triggered by randomly delivering a few drops of water until they spontaneously started seeking water.

Extracellular recordings of Purkinje cell activity were made in awake mice both during spontaneous licking and during motor adaptation. We used quartz-coated platinum/tungsten electrodes ($R = 2$–$5$ M$\Omega$, outer diameter $= 80$ μm, Thomas Recording, Giessen, Germany) randomly placed in an $8 \times 4$ matrix (Thomas Recording), with an inter-electrode distance of 305 μm, and with a minimal depth of 500 μm below the brain surface. The voltage signal was digitized at 24 kHz, using a 1–6000 Hz band-pass filter, 22× pre-amplified and stored using a RZ2 multi-channel workstation (Tucker-Davis Technologies, Alachua, FL, USA).

At the end of the recording session, the electrodes were retracted, and the mouse was anaesthetized again (4% v/v in $O_2$), after which it was killed with cervical dislocation. Death was verified by the absence of breathing.

### Optogenetic stimulation

In a subset of experiments, optogenetic stimulation was applied. To this end, we placed one or two optic fibres (diameter 400 μm, Thorlabs, Newton, NJ, USA) on the brain surface using micromanipulators. Optic fibre placement followed placement of the electrodes, as described above.

With the optic fibres, we could apply blue light (470 nm, 5 mW) for optogenetic stimulation. In a group of four mice, we stimulated vermal Purkinje cells left and right of the midline. In four other mice, we stimulated the right paravermis and the adjacent and more lateral part of lobule VI. We gave pulses with a duration of 160 ms (corresponding to approximately one inter-lick interval)

starting 10 ms after detection of a randomly selected lick. Stimulation of either one or both optic fibres was randomly intermingled. In a third set of experiments, we stimulated only right paravermal Purkinje cells, alternating 160 ms pulses with 80 ms pulses starting 10 ms after lick detection or 80 ms pulses starting 90 ms after lick detection. An extra session was performed with the last group of mice, during which we recorded spikes from Purkinje cells during light stimulation to test its efficacy.

### Analysis of electrophysiological data

Spikes were detected off-line using SpikeTrain (Neurasmus, Rotterdam, The Netherlands). A recording was considered to originate from a single Purkinje cell when it contained both complex spikes (identified by a stereotypic waveform, overshooting and the presence of spikelets) and simple spikes, and in which each complex spike was followed by a pause of at least 8 ms before simple spike firing resumed. We included in our analysis all the recorded cells for which the quality remained constant for at least 20 trials. When looking at complex spike and simple spike modulations, we built peri-stimulus time histograms (PSTHs) using bins of 10 ms. Modulation depth was defined as the difference between maximum and minimum fluctuation in the mean spike modulation to each lick of a bout within a window of 200 ms.

To relate the activity of Purkinje cells to their recording location, we constructed maps of the cerebellar cortex of the entry points of the electrodes. To reduce the impact of unequal sampling, we intrapolated the values, reducing the $8 \times 4$ recording grid to a $7 \times 3$ matrix.

### Phase analysis

The phase values of 0, $\pi$ and $2\pi$ correspond to the moments of protrusion start, maximal protrusion, and end of the retraction, respectively. We considered the values of magnitude-squared coherence and cross spectrum phase between spike-lick cross correlograms and the lick autocorrelograms.

### Model

To develop a model that explains the relationship between Purkinje cell activity and licking behaviour, we adapted an XGBoost classifier model (Chen & Guestrin, 2016). Initially, we processed the raw data on spikes and behaviour, cutting it into 200 ms snippets. These snippets included a 10 ms overlap to ensure continuity. Within these snippets, we calculated three key features: simple spike frequency, simple spike local coefficient of variation (CV2), and complex spike frequency. CV2 was calculated as follows: $2 |ISI_{n+1} - ISI_n|/(ISI_{n+1} + ISI_n)$, where

ISI is inter-spike interval (Shin et al., 2007). The snippets were then grouped based on observed behaviour (presence or absence of licks during a snippet). To create a balanced training set, we randomly selected the same number of snippets from epochs with and without licking. The number was defined by two-thirds of the total amount of the smaller of the two groups. The remainder of the data served as the test set for evaluating our model. For training, we configured the model with 4096 estimators and set the maximum tree depth to 8. We utilized a faster histogram-optimized approximate greedy algorithm for enhancing the gradient boost tree. The performance of the trained models was assessed using the test set to determine the correlation between spike activity and licking behaviour. To elucidate the influence of each feature on the model's output, we employed a game-theoretic method known as SHAP (SHapley Additive exPlanations) (Lundberg et al., 2020). By applying SHAP to our validated models, we were able to estimate the contribution of each feature to the model's predictions, providing insights into the underlying mechanisms linking neural activity to behaviour. For each Purkinje cell and each parameter, the SHAP value was calculated as the mean of the absolute values of the SHAP values per 200 ms window. Purkinje cells were considered to be *predictive* of licking behaviour, if both the accuracy of predicting licking bouts and that of the predicting non-licking bouts were >55%, and *strongly predictive*, if these values were both >65%.

### Statistical analysis

*Z*-scores [(value − mean)/SD] were calculated using the mean and SD of baseline activity. Depending on the analysis, we used different baseline intervals. To calculate spiking modulation around the beginning of licking bouts, the baseline was −1000 to −500 ms before bout onset. For the end of licking bouts, we used as baseline the period from 500 to 1000 ms after bout end. To normalize the spike modulation to licks of different coordinates, we used the period from 1000 to 250 ms prior to the licks of interest.

For analysis of the motor adaptation experiments, lick coordinates were normalized to the coordinates of licks prior to the movement of the lick-port; complex spike modulation to target movements was normalized to a similar 500 ms baseline preceding the beginning of licking bouts as for lick timing modulation; simple spike modulation to licks of interest following target movements was normalized to a similar 750 ms baseline preceding the licks of interest as for the analysis of tongue coordinates during spontaneous licking.

Data were tested for normality using the Kolmogorov–Smirnov test, and when normally distributed Student's paired *t* tests was used, unless

indicated otherwise. Unless stated afterwards, data were summarized as means (SD).

### Use of large language models

The authors did not use large language models, or any artificial intelligence methods when preparing the text and figures, with the exception of the XGBoost classifier model used to analyse the data, as described above.

## Results

Purkinje cells can encode different forms of rhythmic behaviour, like walking, breathing and whisking (Romano et al., 2018, 2020; Sauerbrei et al., 2015). Even though the complex spike activity and the simple spike activity of Purkinje cells have both been shown to correlate with rhythmic licking in separate experiments (Bryant et al., 2010; Cao et al., 2012; Gaffield et al., 2022; Welsh et al., 1995), it has remained unclear to what extent the combined activity of complex spikes and simple spikes of individual Purkinje cells can encode licking.

### Purkinje cell activity relates to spontaneous licking

To study putative interactions between complex spike and simple spike firing during rhythmic licking, we accustomed mice to lick from a water spout in front of their mouth, while being head-fixed in the recording set-up. Under these conditions, mice licked rhythmically in bouts with lick frequencies generally between 4 and 9 Hz (Fig. 1*A*), similar to what can be observed in freely moving mice (Boughter et al., 2007). During sessions with spontaneous licking, we recorded the activity of Purkinje cells in lobules VI and VII of the vermis and the adjacent hemispheric lobules crus 1 and crus 2. In many, but not all, Purkinje cells, we observed rhythmic modulation of complex spike and simple spike activity along with rhythmic licking. More specifically, when we analysed the recordings of the entire periods of licking, in 21 out of 84 (25%) recorded Purkinje cells, complex spike modulation was considered statistically significant ($Z > 3$), while statistically significant simple spike modulation occurred in 58 out of 84 (69%) Purkinje cells. In total 19 (23%) of the recorded Purkinje cells showed modulation of both their complex spikes and their simple spikes (Fig. 1*B–D*).

### Complex spikes signal behavioural state changes

In view of the relatively low fraction of the Purkinje cells showing simple spike modulation that also displayed complex spike modulation during rhythmic licking, we wondered whether we might have missed specific contributions of complex spikes. As complex spike firing

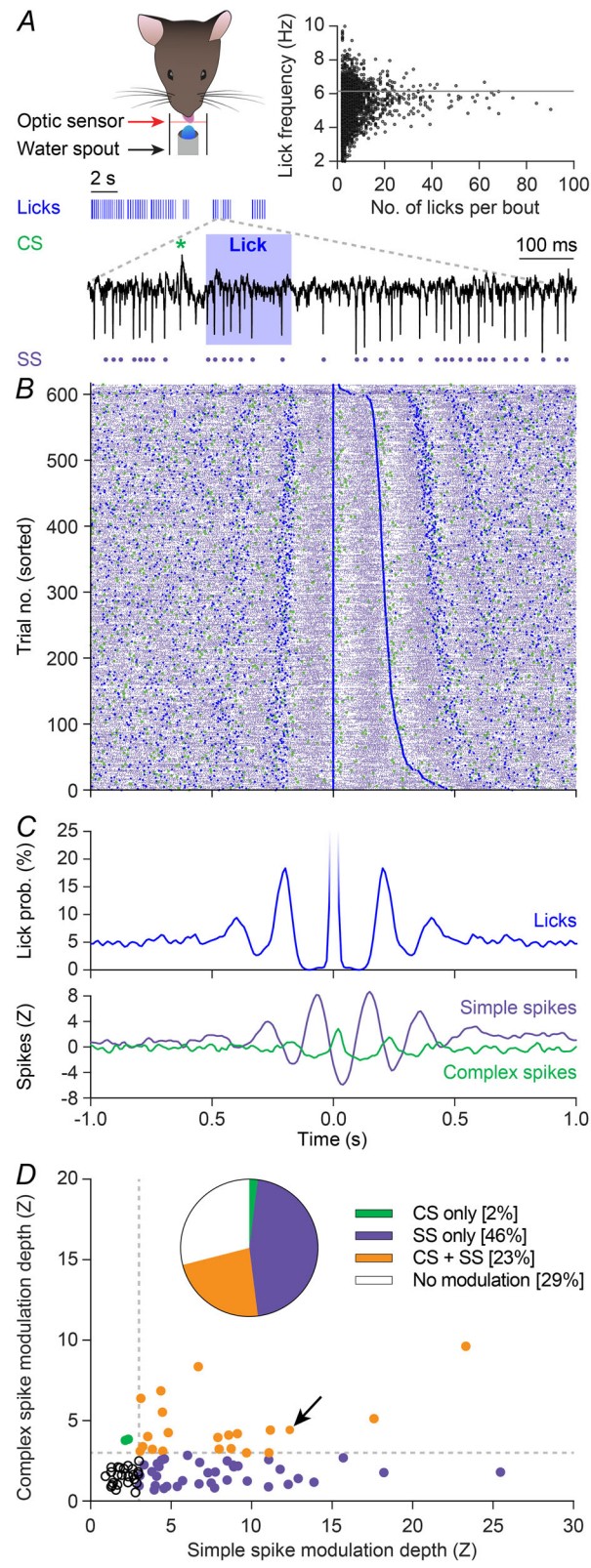

Purkinje cells, we made extracellular recordings revealing the timing of complex spikes (green asterisk) and simple spikes (purple dots). *B*, raster plot showing the distribution of licks (blue), complex spikes (green) and simple spikes (purple) during one session with spontaneous licking. Trials are sorted based on the duration of the inter-lick interval. *C*, peri-stimulus time histograms (PSTHs) from the licks (top) and spikes (bottom) from the session shown in *B*. Notice that all three PSTHs display oscillations at similar frequencies. *D*, in individual Purkinje cells, the modulation depths of complex spikes and simple spikes were not necessarily correlated, with most Purkinje cells showing a stronger modulation of their simple spikes than of their complex spikes. Data from 84 Purkinje cells in 32 mice. The arrow indicates the example cell shown in *A–C*. CS, complex spike; SS, simple spike. [Colour figure can be viewed at wileyonlinelibrary.com]

has been associated with changes in behavioural state (Streng et al., 2017, 2022; Wagner et al., 2021), we next focused on the start and end of licking bouts. In the 300 ms interval centred on the detection of the first lick of a bout, we observed a statistically significant increase ($Z > 3$) of complex spike firing in 33 out of 84 (39%) recorded Purkinje cells (Fig. 2*A–E*). Similarly, the end of a licking bout corresponded with increased complex spike firing in 43 (51%) Purkinje cells. Although more Purkinje cells were active around the end than around the start of a licking bout, this difference was not statistically significant ($P = 0.1628$, Fisher's exact test, Fig. 2*E*).

Of the 58 Purkinje cells that showed a significant complex spike response, 18 (31%) signalled both the start and the end of licking bouts with their complex spikes; the others signalled either the start or the end. The notion that the Purkinje cells signalling the start of a licking bout are not necessarily the same as those signalling the end of a licking bout was underscored by the lack of a correlation between the amplitudes of complex spike modulation between both state changes ($r = 0.0147$, $P = 0.8943$, $n = 84$ Purkinje cells, Spearman's rank correlation test, Fig. 2*E*).

Simple spike modulation, in contrast, was more related to single licks (Fig. 1), and as a consequence simple spike modulation around the first and last lick of a bout were strongly correlated ($r = 0.4639$, $P < 0.0001$, $n = 84$ Purkinje cells, Spearman's rank correlation test, Fig. 2*F*). Thus, complex spike firing was more related to changes in the behavioural state, i.e. starting or ending a licking bout, while simple spike firing was more related to the execution of individual licks.

## Decoding of Purkinje cell activity patterns with machine learning inference

The Purkinje cells of crus 1 and surrounding areas of the cerebellar cortex are involved not only in licking, but also in myriad other related behaviours (Bina et al., 2021; Bosman et al., 2010; Gaffield et al., 2022; Heffley & Hull,

**Figure 1. Purkinje cells encode rhythmic licking**
*A*, mice make repetitive tongue movements when licking, and these licks are organized in bouts. In the experimental set-up, the median lick frequency of all bouts was just above 6 Hz (horizontal line in inset). To correlate tongue movements to activity of cerebellar

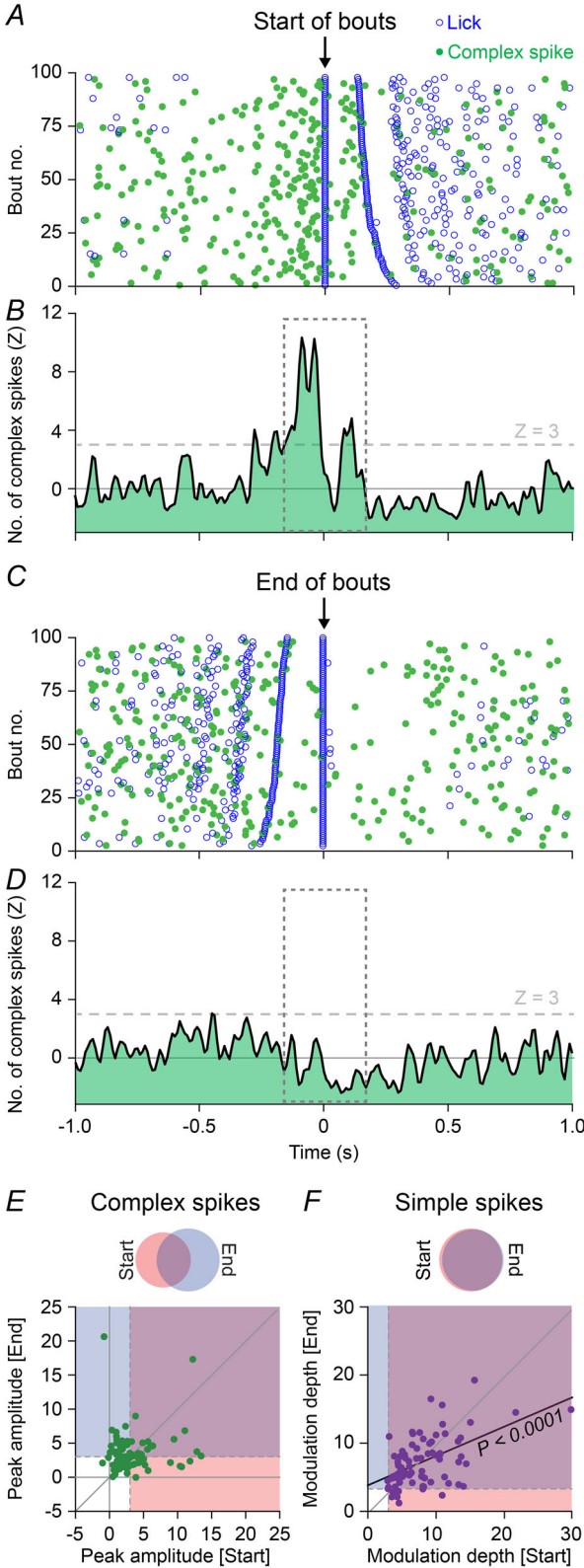

the start of licking bouts. *C*, the distribution of licks and complex spikes around the end of licking bouts, with the bouts sorted on the duration of the last inter-lick interval. This particular Purkinje cell did not show increased complex spike firing at the end of licking bouts. *D*, complex spike PSTH around the end of licking bouts. *E*, the peak amplitudes of complex spike modulation around the start *vs*. the end of licking bouts (see dashed intervals in *B* and *D*) showed no statistically significant correlation ($r = 0.0147$, $P = 0.8943$, $n = 84$ Purkinje cells in 32 mice, Spearman's rank correlation test). Venn diagram indicates the relative fractions of Purkinje cells modulating complex spike firing at the start or end of licking bouts, or during both start and end. *F*, same as *E*, but for simple spike modulation depth. All recorded cells modulated significantly around either the start or end of licking bouts, and the modulation depths around the start and end of licking bouts were highly correlated per Purkinje cell ($r = 0.4639$, $P < 0.001$, $n = 84$ Purkinje cells in 32 mice, Spearman's rank correlation test). Venn diagram indicates the relative fractions of Purkinje cells modulating simple spike firing at the start or end of licking bouts, or during both start and end. [Colour figure can be viewed at wileyonlinelibrary.com]

**Figure 2. Complex spikes signal behavioural state changes**
*A*, the distribution of licks and complex spikes around the start of licking bouts during one session with spontaneous licking. The bouts are sorted based on the duration of the first inter-lick interval. *A–D* are from the same recording session. *B*, complex spike PSTH around

2019; Hoang et al., 2023; King et al., 2019; Kitazawa et al., 1998; Lackey et al., 2024; Romano et al., 2020; Streng et al., 2018; Wagner et al., 2021). To investigate whether the observed alterations in Purkinje cell spiking patterns are specific for licking behaviour or whether they also occur during periods without licking, we implemented a machine learning workflow. First, using a 200 ms sliding window, we computed for each recorded Purkinje cell three key features: complex spike frequency, simple spike frequency and simple spike irregularity (CV2). These features were then grouped, based on the presence or absence of licking during each of the 200 ms intervals. An adapted version of the XGBoost classifier model (Chen & Guestrin, 2016) was trained for each cell to detect the relationship between cell activity and licking behaviour.

In 58 out of 84 (69%) of the recorded Purkinje cells we observed a prediction accuracy of >55%, and in 29 (35%) Purkinje cells the prediction accuracy exceeded 65% (Fig. 3*C*). The prediction values for licking bouts and inter-bout intervals were correlated ($r = 0.6600$, $P < 0.0001$, Pearson's correlation), but there was a tendency towards a higher accuracy for bout *vs*. inter-bout intervals in the Purkinje cells with a high predictive value (accuracy >65%: $P = 0.0245$, paired *t* test, Fig. 3*C*).

To understand which of the three key features (complex spike frequency, simple spike frequency or simple spike CV2) had the strongest predictive value, we employed the SHAP method (Lundberg et al., 2020) on the 58 Purkinje cells that showed a prediction accuracy of at least 55% (Fig. 3*A–D*). The absolute SHAP values of simple spike frequency and CV2 were unrelated ($r = 0.1428$, $P = 0.1949$, Spearman's rank correlation, Fig. 3*D*), indicating a heterogeneity in the simple spike encoding of licking. For the Purkinje cells that had a poor-to-moderate predictive power (accuracy

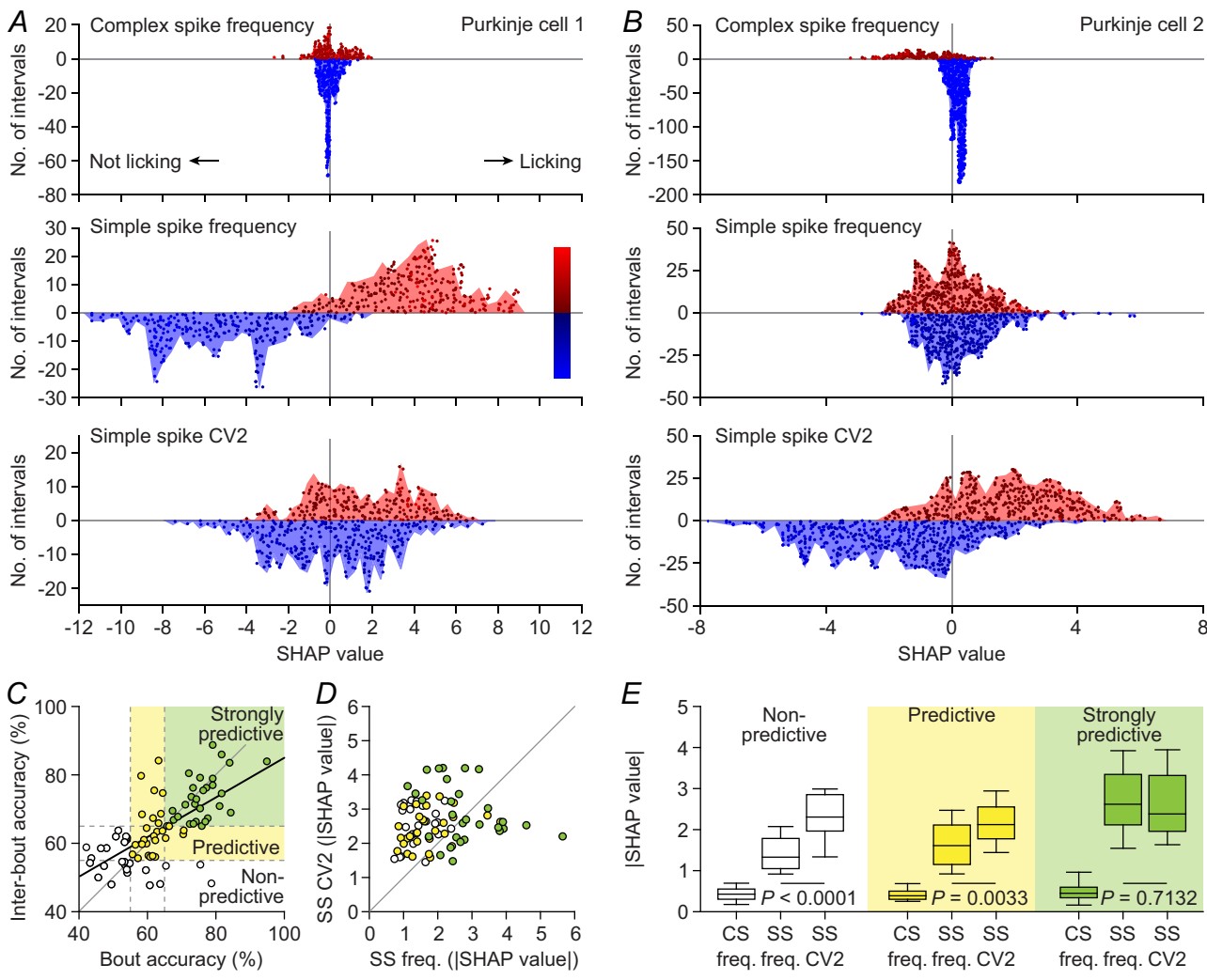

**Figure 3. Decoding of Purkinje cell activity patterns with machine learning inference**
*A*, we adapted an XGBoost classifier model that could explain the relationship between Purkinje cell activity and licking behaviour. Our model based its prediction on three features extracted from the dataset after dividing it in sliding windows of 200 ms (with 10 ms overlap): complex spike frequency, simple spike frequency and simple spike CV2. These features are colour-coded based on their values, and for clarity low values (blue shades) are plotted negatively and high values (red shades) positively. Each snippet is plotted individually, and the blue/red contours display a histogram with the distribution of the snippets. The illustrated data come from a Purkinje cell that particularly showed alterations in simple spike frequency. *B*, same as in *A*, but for a different Purkinje cell in which simple spike CV2 had a higher predictive value than simple spike frequency. *C*, comparison between bout and inter-bout accuracy for each of the 84 Purkinje cells recorded in 32 mice (*r* = 0.6600, *P* < 0.0001, Pearson's correlation test). We discriminate between non-predictive (accuracy <55%), predictive (accuracy between 55 and 65%) and strongly predictive Purkinje cells (accuracy >65%). *D*, the absolute SHAP values for the simple-spike-related features in the 58 lick-related Purkinje cells; these did not show any correlation (*r* = 0.1428, *P* = 0.1949, Spearman's rank correlation test). Symbols are colour-coded according to the accuracy of the Purkinje cells (see *C*). *E*, absolute SHAP values for the three features reveal much lower impact of the complex spike frequency than of the simple spike parameters for the model prediction (*P* < 0.0001, Friedman's ANOVA). *Post hoc* tests revealed that the simple spike CV2 had higher absolute SHAP values than simple spike frequency in Purkinje cells with an accuracy < 65%, while these values were similar in strongly predictive Purkinje cells. CS, complex spike; SS, simple spike. [Colour figure can be viewed at wileyonlinelibrary.com]

<65%), the SHAP values for CV2 were generally larger than for frequency (Fig. 3*D* and *E*). In contrast, in some of the strongly predictive Purkinje cells (accuracy >65%) frequency modulation was more prominent (Fig. 3*D* and *E*). In view of the low frequency of complex spike firing, it was not surprising to find little predictive value of complex spike firing per 200 ms interval (Fig. 3*E*).

## Complex spikes and simple spikes cover the whole of the lick cycle

Given that our previous analyses confirmed modulation of complex spikes as well as simple spikes during licking, we subsequently studied the phase relationships between Purkinje cell activity and licking. To this end, we made a phase transform of all spike times and subsequently performed, for each Purkinje cell, a coherence analysis between licking and spiking. As the phase transform annihilated timing differences between individual licks, this phase-based coherence analysis proved more sensitive than the PSTHs based on timing alone (Fig. 1*D*): 61 out of 84 (73%) recorded Purkinje cells showed a coherence level of more than 0.5 between complex spike firing and licking (Fig. 4*A*). The complex spike activity of each individual Purkinje cell had a preferential phase within the licking cycle, but the preferred phases of the different cells were heterogeneously and relatively randomly distributed (Fig. 4*A*). Together they covered the entire cycle, even though there were slight preferences for the onset and peak of the protrusion. We next plotted the coherence levels for the complex spike modulations of each Purkinje cell based on the insertion points of the electrodes at the surface of the cerebellar hemispheres and vermis. This structure–function analysis did not reveal specific anatomical hotspots (mean coherence medial 0.65 (0.08), lateral 0.68 (0.09), $P = 0.7962$, $n = 9$ locations per side, Mann–Whitney test, Fig. 4*B*).

The coherence analysis showed that the phase relation between simple spike firing and licking was stronger than that for the complex spikes (Fig. 4*C*). In total 77 out of 84 (92%) Purkinje cells showed >0.5 coherence with licking. Moreover, when we spatially plotted the coherence levels for the simple spike modulations, specific anatomical hotspots emerged, with the strongest levels of coherences localized at the border regions between hemispheres and vermis (median coherence medial 0.88 (inter-quartile range (IQR) 0.02), lateral 0.79 (IQR 0.06), $P = 0.0002$, $n = 9$ locations per side, Mann–Whitney test, Fig. 4*D*).

This analysis revealed that most Purkinje cells (56 out of 84; 67%) showed both coherence between complex spikes and licking, as well as between simple spikes and licking (Fig. 4*E*). Instead, just 6% (5 out of 84) of the Purkinje cells displayed coherence with the licking for their complex spikes only, and 25% (21 out of 84) for simple spikes only (leaving only 2 cells with no clear coherence). The phase relations between complex spikes and simple spikes varied strongly between Purkinje cells. In general, though, the majority of Purkinje cells had a tendency to fire complex spikes and simple spikes in anti-phase (Fig. 4*F*).

## Are Purkinje cells also rhythmically active at rest?

The presence of coherence between licking and spiking substantiates the presence of rhythmic firing during licking bouts. Given that Purkinje cells are often involved in different types of orofacial behaviour (Cao et al., 2012; Romano et al., 2020), and that these behaviours typically show similar dynamics (Moore et al., 2013), we wondered whether rhythmic Purkinje cell activity could also be found during periods without licking activity. Hence, we constructed auto-correlograms of simple spike activity during and in between licking bouts. This analysis confirmed rhythmic simple spike activity during licking, but little to no rhythmic activity in the absence of licking (power of simple spike autocorrelograms had a median of 2.4 (IQR 1.9) during lick bouts *vs.* a power of 1.2 (IQR 1.0) in between lick bouts; $P < 0.0001$, Wilcoxon's matched-pairs test; $r = 0.6064$, $P < 0.0001$, Spearman's rank correlation test, Fig. 5). Thus, we conclude that the occurrence of rhythmic simple spike firing was largely confined to periods with licking.

## Purkinje cell activity correlates with the trajectory of the tongue

After establishing that Purkinje cell simple spike firing can be linked to rhythmic licking, we wondered to what extent simple spike firing affects tongue movements. To this end, we analysed video recordings of spontaneous licking to monitor the variation in tongue movements between trials. For each mouse, we plotted the locations of maximal protrusion of each lick, and divided the distribution of end-points in tertiles, both on the rostro-caudal and on the left–right axis (Fig. 6*A*). For each one of the nine resulting areas, we calculated the mean normalized simple spike firing frequency preceding the end point of each lick at two time intervals, the first one from 150 to 75 ms before the end point and the second one from 75 to 0 ms before this moment of maximal protrusion (Fig. 6*B*).

Next, we quantified the differences in the simple spike firing rate during tongue protrusion (75 to 0 ms before reaching the maximal protrusion) of licks inclined to the right or the left side. More simple spikes were fired on average in this 75 ms interval during right-bending (ipsiversive) licks (mean simple spike frequency ($Z$): ipsiversive 1.24 (2.54) *vs.* contraversive 0.34 (2.22), $P = 0.0205$, paired *t* test, Fig. 6*C*). Analogously, the peak values of the two distributions also differed (peak simple spike frequency ($Z$): ipsiversive 4.01 (2.52) *vs.* contraversive 2.94 (2.21), $P = 0.0131$, paired *t* test). The contrast between ipsi- and contraversive licks was emphasized by the absence of a significant correlation in maximal simple spike firing during both conditions ($r = 0.2206$, $P = 0.1199$, Spearman's rank correlation test, Fig. 6*D*).

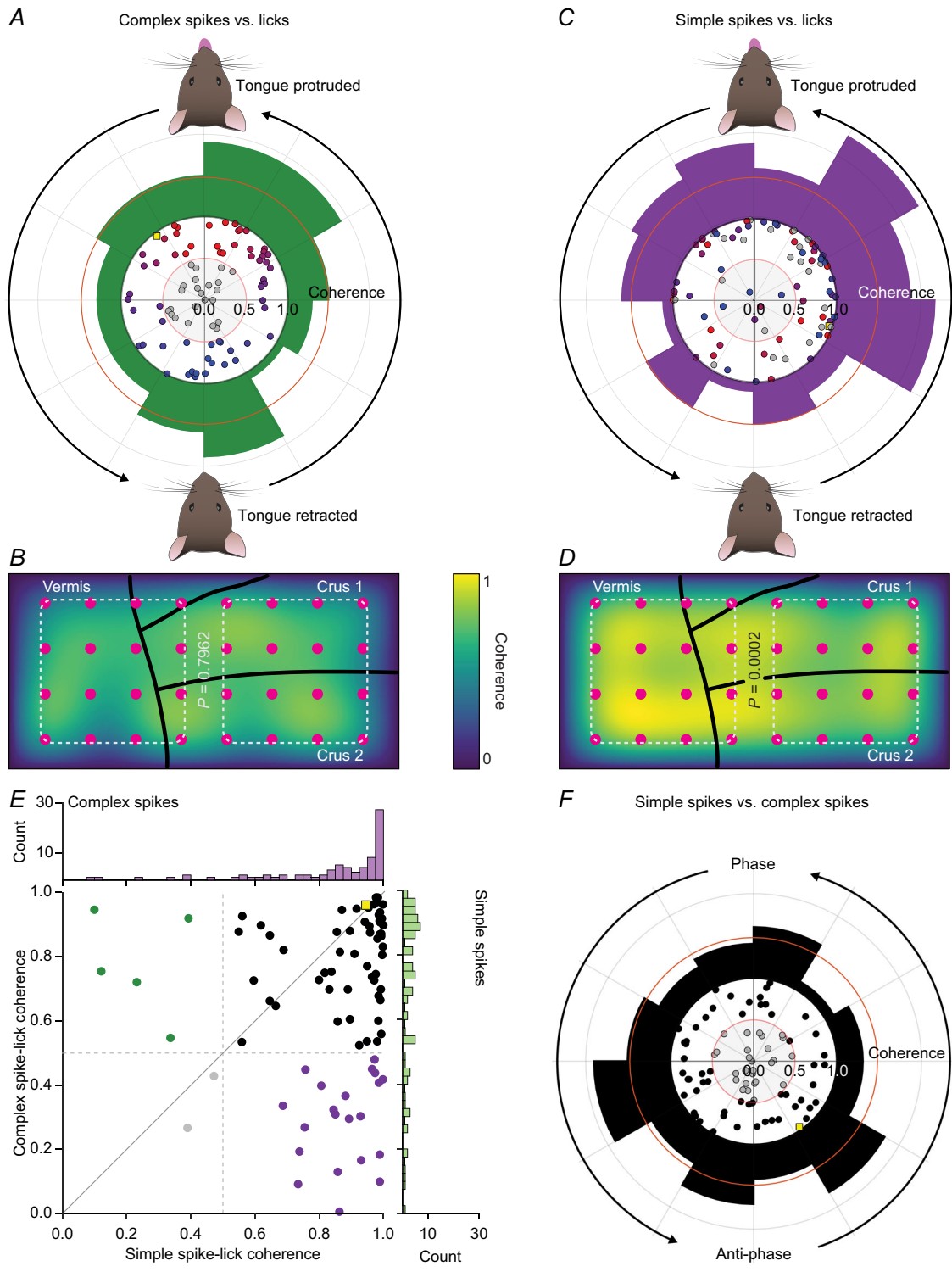

**Figure 4. Coherent activity between complex spikes, simple spikes and licking**

*A*, complex spike distributions around licks show coherence values >0.5 in 61 out of 84 recorded Purkinje cells. The preferential phase in which complex spikes occur in those cells varies along the licking cycle. Each dot represents a cell and is colour-coded by its preferential phase during rhythmic tongue movements. The yellow square represents the example cell shown in Fig. 1 (notice that high coherence value and location in the third quadrant reflect the oscillating behaviour and a main peak that follows the tongue detection, respectively, as observed for the complex spike PSTH from Fig. 1D). *B*, heatmap showing the mean coherence level for complex spikes in the recording area

based on the points of insertion of the recording electrodes on the surface of the cerebellum (pink circles). The mean coherence between complex spike firing and licking was similar between the medial and lateral part of the recording area ($P = 0.7962$, $n = 84$ Purkinje cells from 32 mice, Mann–Whitney test). C, same plot as in A, but for simple spikes. D, same plot as in B, but for simple spikes. The mean coherence between simple spike firing and licking was stronger in the medial than in the lateral part of the recording area ($P = 0.0002$, $n = 84$ Purkinje cells from 32 mice, Mann–Whitney test). E, scatter plot showing the lack of a correlation between complex spike-lick coherence and simple spike-lick coherence, independent from the preferred phase. Some cells show only complex spike-lick coherence (green), others only simple spike-lick coherence (purple). F, coherence of complex spike *vs.* simple spike firing. [Colour figure can be viewed at wileyonlinelibrary.com]

This suggests that simple spikes may enhance tongue movements towards the ipsilateral side.

We next investigated whether complex spikes could also play a role in the laterality of tongue movements, as complex spikes are followed by a pause in simple spike firing. Trials with a complex spike displayed fewer simple spikes (mean Z-score of the 150 ms interval before lick detection: without complex spikes: 0.92 (1.40); with complex spikes −0.69 (1.37), $P < 0.0001$, paired *t* test, $n = 40$ Purkinje cells, Fig. 6E). This did not result in a lateral bias of tongue movements (−0.01 (0.27) *vs.* 0.01 (0.29) mm; $P = 0.4312$, paired *t* test), but rather in a reduction in the extension of tongue protrusion (4.18 (0.88) *vs.* 4.15 (0.89) mm; $P = 0.0013$, paired *t* test, Fig. 6F).

### Adaptation of Purkinje cell activity during targeted tongue movements

Next, we tested the ability of mice to adapt their tongue movements to a displaced target (Fig. 7A). To this end, we moved the lick-port 3 mm to the right during the retraction phase of randomly selected licks. The lick-port stayed at that position during 750 ms and then returned to the central position. While the first lick after lick-port displacement did not yet show statistically significant bending, the second and third licks were adapted to the new target position ($P < 0.0001$, Friedman's ANOVA with *post hoc* tests (based on tongue position in mm) *vs.* lick 0: lick 1: $P = 0.5683$, lick 2: $P = 0.0028$, lick 3: $P < 0.0001$, $n = 13$ mice, Fig. 7B). The movement back to the centre had a very similar impact on the tongue position ($P < 0.0001$, Friedman's ANOVA with *post hoc* tests *vs.* lick 0: lick 1: $P = 1.0000$, lick 2: $P = 0.0005$, lick 3: $P < 0.0001$, $n = 13$ mice, Fig. 7B).

The movement of the lick-port was associated with changes in simple spike firing (peak in simple spike firing around tongue protraction for lick 0: $P = 0.5347$, lick 1: $P = 0.0428$, lick 2: $P = 0.0204$, lick 3: $P = 0.0018$, Wilcoxon's matched-pairs test with the activity for licks 2 and 3 remaining significantly different after Benjamini–Hochberg correction for multiple comparisons; Fig. 7C). This correlation between increased simple spike firing and ipsilateral bending of the tongue is in line with our findings during spontaneous licking (Fig. 6).

Movement of the lick-port triggered increased complex firing in 27 (42%) out of 65 Purkinje cells (Fig. 7D). In these responsive Purkinje cells, statistically significant complex spike responses could be triggered by rightward movements (ipsilateral relative to the recording position) in 21 Purkinje cells and by movements back to the centre in 11 Purkinje cells. Of these Purkinje cells, five showed significant modulation in both directions, the other 22 in either one direction (Fig. 7E). The rightward (i.e. ipsiversive) movements triggered stronger complex spike responses in the 27 responsive Purkinje cells ($P < 0.0001$, Wilcoxon's matched-pairs test, Fig. 7E). Thus, unexpected movements of the target for the tongue movements triggered complex spike responses, and these responses were typically direction-sensitive. Overall, the complex spike responses were anti-correlated with the changes in simple spikes ($r = -0.3316$, $P = 0.0070$, Spearman's rank correlation test. Fig. 7F). Thus, complex spikes signalled the sensory event of the moving lick-port, while the simple spikes correlated with the trajectory of the altered tongue movements.

### Purkinje cell stimulation results in bending of the tongue

Increased simple spike firing correlated with ipsilateral bending of the tongue during protrusion. To find out if there is a causal relation between simple spike firing and tongue trajectory, we used optogenetic stimulation of Purkinje cells to evoke increased simple spike firing (Lindeman et al., 2021; Witter et al., 2013). We used randomly selected licks to trigger optogenetic stimulation in phase with licking. An optic fibre of 400 μm diameter was placed on the surface of the cerebellum at the border between crus 1 and crus 2, just right of the vermis (Fig. 8A and B), as this is the area with the strongest coherence between simple spike activity and licking (Fig. 4D).

In general, optogenetic stimulation triggered a robust increase in simple spikes, which was followed by increased complex spike firing approximately 70 ms after the end of stimulation (Fig. 8C–E), as observed previously (Witter et al., 2013). We tested the impact of stimulation either during tongue retraction of the detected lick, tongue protrusion of the next lick, or using a twice as long

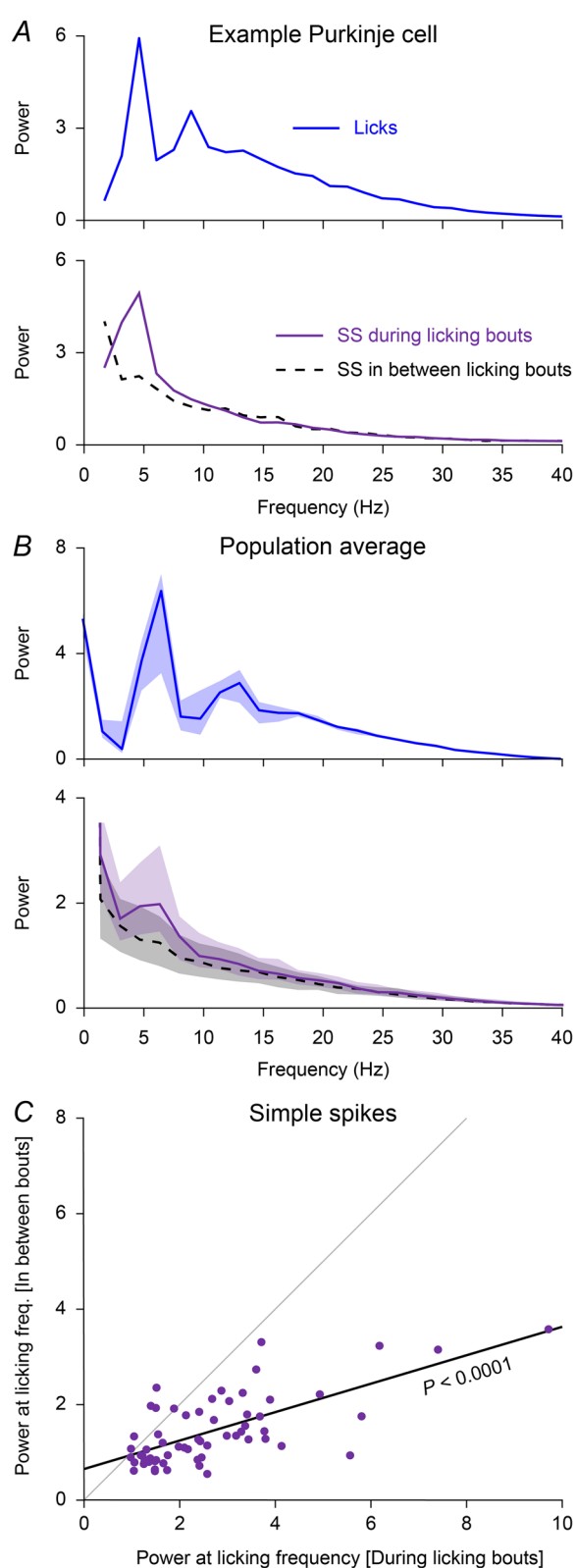

5 Hz. Bottom: rhythmic firing at the same frequency was also evident in the autocorrelogram of simple spikes (SS) fired during, but not in between, licking bouts. *B*, power spectra of the autocorrelograms of licking (top) and of simple spike firing (bottom) for the 61 out of 84 recorded Purkinje cells that showed an evident peak around the licking frequency. Medians with shades representing inter-quartile range (IQR). *C*, for each of the 61 Purkinje cell included in the analysis of *B*, the power at the preferred licking frequency during that recording during *vs.* in between licking bouts. The power was typically stronger during licking bouts ($P < 0.0001$, Wilcoxon's matched-pairs test), yet correlated per Purkinje cell ($r = 0.6064$, $P < 0.0001$, Spearman's rank correlation test). [Colour figure can be viewed at wileyonlinelibrary.com]

interval (160 instead of 80 ms) to cover both (for example, see Fig. 8*B*). Optogenetic stimulation resulted in altered tongue movements: the first lick occurring after stimulation onset was bent in general ipsilaterally (*x*-coordinate lick 1 *vs.* lick 0, 10–90 ms: $P = 0.0168$; 90—170 ms: $P < 0.0001$; 10–170 ms: $P < 0.0001$, paired *t* tests with Bonferroni correction, $n = 7$ mice, Fig. 8*F*–*K*) indicating that the simple spike rate can indeed affect the tongue trajectory. Optogenetic stimulation also led to shorter tongue protrusions, with the effect of early stimulation (lick 1: $P = 0.0068$, paired *t* test, Fig. 8*I*) visible earlier on than during late stimulation (lick 2: $P = 0.0189$, paired *t* test, Fig. 8*J*), which may be explained by the occurrence of increased complex spike firing around 100 ms after the start of optogenetic stimulation (in line with Fig. 6*F*).

In another batch of mice, we introduced a second optic fibre at the opposite hemisphere (Fig. 9*A*), and stimulated in random fashion either one side at a time or both simultaneously, using the longer stimulus interval of the previous experiment (10–170 ms after lick detection). Stimulating on the right side induced, as before, a right-ward bending of the next tongue protrusion, while stimulating on the left caused a left-ward bending (lick 1 following Purkinje cell stimulation on the left *vs.* on the right, $P < 0.0001$, $n = 4$ mice, paired *t* test, Fig. 9*B* and *C*). This indicates that enhancing simple spike firing, thus inhibiting the cerebellar nuclei, could suppress the contraction of ipsilateral muscles.

With a last group of mice, we investigated the impact of more lateral stimulations and found that they were significantly less effective in generating an ipsilateral bias in tongue movements (lick 1 following Purkinje cell medial *vs.* lateral stimulation, $P = 0.0220$, $n = 4$ mice, paired *t* test, Fig. 9*D*). Thus, artificially increasing simple spike firing of specific sets of Purkinje cells in phase with licking could affect the execution of individual cycles of tongue movements during licking, proving the causal relation between Purkinje cells output and directionality of tongue movements.

**Figure 5. Oscillations in phase with licking are absent outside of licking bouts**

*A*, top: power spectrum calculated from the autocorrelogram of licks during a randomly selected session of spontaneous licking. During this session, licking was rhythmic at a frequency of approximately

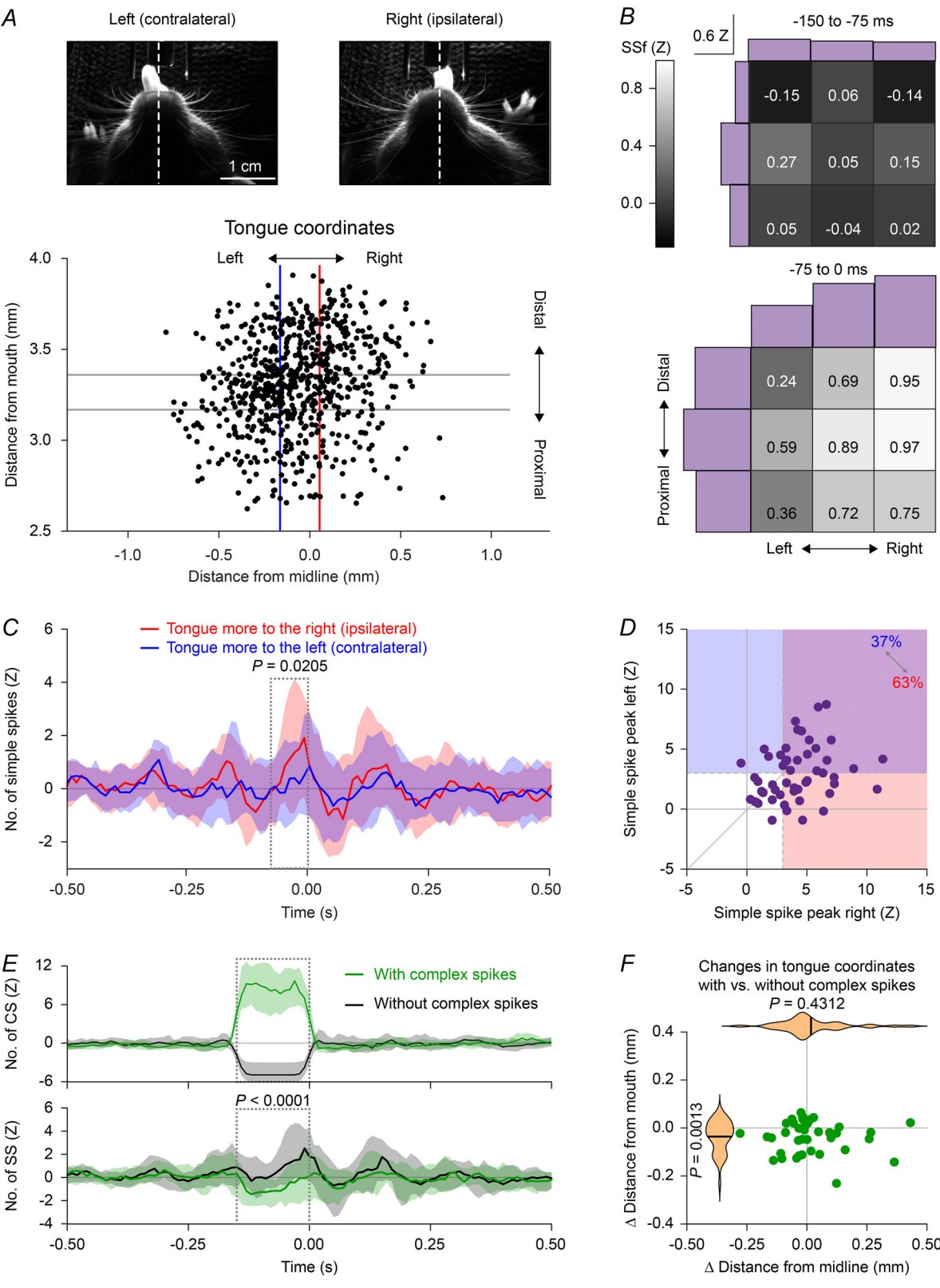

**Figure 6. Purkinje cells encode the coordinates of tongue protrusion**
*A*, using video analysis, licks were classified based on the coordinates of the tongue at the moment of maximal protrusion. Two example video frames for licks to the left and to the right are shown. For the whole session, the

rosette resulting from the tongue coordinates is divided in tertiles on the distal–proximal and left–right axes. Note that this mouse had a bias towards the left, and our analysis adapted to this individual bias. *B*, heatmaps showing the mean simple spike frequency (in *Z* scores) preceding licks in two consecutive time intervals relative to the moment of maximal protrusion (top and bottom) for the nine spots obtained after the subdivision of each rosette. Histograms of the spike frequencies along the two axes are shown in purple. SSf, simple spike frequency. *C*, median simple spike PSTHs showing increased simple spike firing during tongue protraction (between 75 and 0 ms prior to reaching the maximal protraction) of licks bending ipsilaterally *vs.* contralaterally: $P = 0.0205$, $n = 84$ Purkinje cells in 32 mice, paired *t* test. Medians with shades representing IQR. *D*, scatter plot of the peaks in simple spike firing during the interval highlighted in *C*. $r = 0.2206$, $P = 0.1199$, $n = 84$ Purkinje cells, Spearman's rank correlation test. *E*, PSTHs showing the distribution of complex spikes (CS, top) and simple spikes (SS, bottom) for licks where complex spikes occurred or not in a time window of 75 ms preceding each lick. $P < 0.0001$, $n = 84$ Purkinje cells, paired *t* test. Medians with shades representing IQR. *F*, tongue protrusion of licks preceded by complex spikes was shorter, but did not have a lateral bias (violin plots: black lines indicate mean). Tongue extension (*y*-axis): $P = 0.0013$; bending (*x*-axis) $P = 0.4312$, $n = 84$ Purkinje cells, paired *t* tests. [Colour figure can be viewed at wileyonlinelibrary.com]

## Discussion

Mice, like most mammals, make fast rhythmic tongue movements when drinking. This rhythm is set by a central pattern generator in the brainstem (Brozek et al., 1996; Dempsey et al., 2021; Kleinfeld et al., 2023; Travers et al., 1997), while we show here that tongue movements can be modulated by cerebellar Purkinje cells. Complex spikes of Purkinje cells can be related to behavioural state changes and signal unexpected movement of the target position of the tongue, while their simple spikes are associated with the (a)symmetry of tongue movements. By challenging mice to adjust their tongue movements towards a quickly moving target, we could demonstrate that complex spike signals are modulated proportionally to the error in the direction of the movement and that alterations in simple spike firing are consistent with the altered movements themselves. Accordingly, optogenetic stimulation of Purkinje cell activity results in changes in the tongue trajectory. Thus, the activity of Purkinje cells not only reflects the way goal-directed tongue movements are executed, but can actually alter these.

### Cerebellar activity during licking

During rhythmic licking, related simple spike firing was observed abundantly in the lateral cerebellum, in line with previous results (Bryant et al., 2010; Cao et al., 2012; Gaffield et al., 2022). We found that the majority of cells that modulate rhythmically are located medially within the hemispheres. Each Purkinje cell has a pre-ferred phase lag compared to tongue movement, so that together Purkinje cells cover the whole lick cycle, similar to other rhythmic behaviours like walking (Sauerbrei et al., 2015) and respiration (Romano et al., 2020). As during respiration (Romano et al., 2020), there is an uneven distribution of preferred phases of individual Purkinje cells over the licking cycle. Relatively many Purkinje cells are particularly active during the later phases of tongue protrusion. Increased simple spike firing during this period correlates with an ipsiversive bending of the

tongue. Purkinje cell output controls the activity in the cerebellar nuclei, which in turn modify activity of the neurons downstream in the reticular formation that are engaged in licking (Lu et al., 2013).

Complex spike firing was also modulated in a coherent fashion with spontaneous rhythmic licking, as reported previously (Welsh et al., 1995), but the related modulation depth was less than that of simple spikes, and the number of Purkinje cells found with a significant complex spike modulation during rhythmic licking was lower than that for simple spikes. This is in line with the notion that simple spikes are more relevant for acute motor output than complex spikes (Bina et al., 2021; Chen et al., 2016). Given the low frequency of complex spike firing and the inability to change this over longer periods (Ju et al., 2019; Negrello et al., 2019), the predictive value of the presence or absence of a complex spike during a 200 ms inter-val for the acute behaviour is limited, as demonstrated in Fig. 3. As discussed below, the role of the complex spikes in coordination of skilled tongue movements appears more prominent during the adaptation of licking movements.

### Interaction between complex spikes and simple spikes

Purkinje cells are unique in producing two different types of action potentials: complex spikes and simple spikes. Complex spikes are relatively rare: they occur with a frequency of around 1 Hz and thereby form only 1–3% of Purkinje cell spikes (Negrello et al., 2019; Zhou et al., 2014). They are exclusively evoked by activity of the one or two climbing fibres that innervate each adult Purkinje cell, and they are associated with a strong increase in intracellular $Ca^{2+}$ in Purkinje cells (Bosman et al., 2008; Busch & Hansel, 2023; De Zeeuw et al., 2011). The much more frequent simple spikes are generated intrinsically by Purkinje cells, while their frequency can be increased by parallel fibres or decreased by molecular layer interneurons (Cerminara et al., 2015; De Zeeuw et al., 2011). The strength of parallel fibre-to-Purkinje

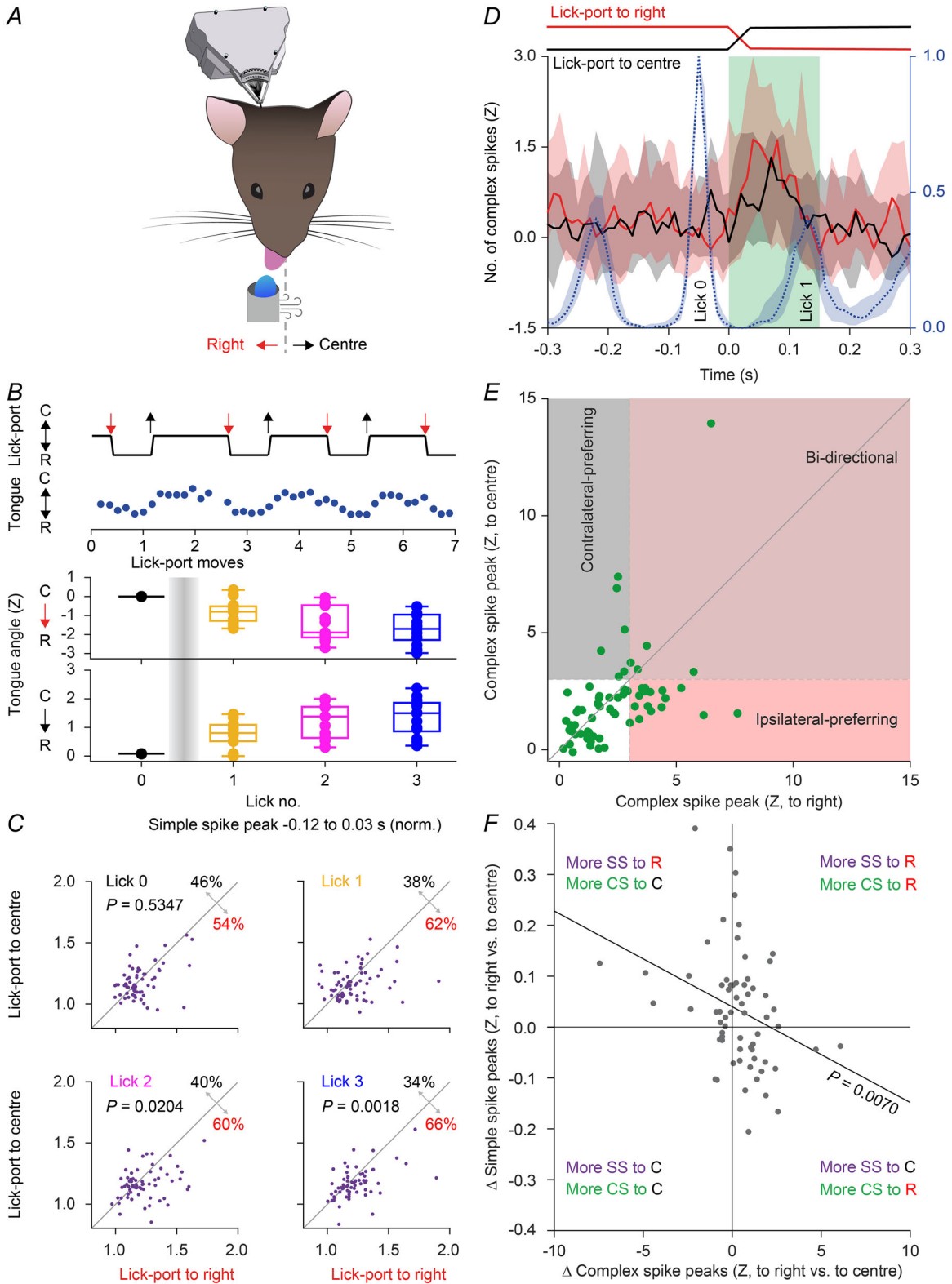

**Figure 7. Purkinje cell firing reflects motor adaptation**

*A*, experimental scheme: 3 mm rightward movements of the lick-port occurring during the retraction phase of the tongue were triggered on randomly selected licks. The lick-port was moved back to the original position after 750 ms and stayed in the central position for at least 750 ms, so that the direction of the 4–5 licks following each movement needed to be adjusted for a more efficient consumption of the water. Outside of the 1.5 s trial time,

some licks were not affected by any lick-port movement for a minimum of 300 ms and used as baseline. *B*, a randomly selected epoch showing that left–right changes in position of the lick-port lead to left–right changes in the endpoints of tongue protrusions. Mice adapted the direction of the tongue protrusions partially for the first lick following the trials, fully for the second and third licks. $P < 0.0001$, Friedman's ANOVA, with *post hoc* tests comparing the most extended position (in mm) of the tongue between licks 1–3 *vs*. lick 0 (rightward movements: lick 1: $P = 0.5683$, lick 2: $P = 0.0028$, lick 3: $P < 0.0001$; centreward movements: lick 1: $P = 1.0000$, lick 2: $P = 0.0005$, lick 3: $P < 0.0001$, $n = 13$ mice). *C*, scatterplots of the peak values of the simple spike distributions around licks of interest for lick-port movements to the right and back (rightward *vs*. centreward movements, lick 0: $P = 0.5347$, lick 1: $P = 0.0428$, lick 2: $P = 0.0204$, lick 3: $P = 0.0018$, Wilcoxon's matched-pairs test with licks 2 and 3 statistically significant after Benjamini–Hochberg correction for multiple testing). *D*, peri-stimulus time histograms showing complex spike responses elicited after rightward (red) or centreward (black) movements of the lick-port (27/65 cells with a peak response larger than 3 SD). Complex spike responses typically preceded the moment of maximum protrusion of the tongue of the subsequent lick (see lick distribution in blue). Medians with shades representing IQR. *E*, the majority of Purkinje cells (63%) displayed a stronger complex spike response when the lick-port moved in the ipsilateral direction (to the right). *F*, correlation between the difference in selectivity (right − centre) for complex spike peaks and the mean simple spike peaks around licks 2 and 3 ($r = -0.3316$, $P = 0.0070$, Spearman's rank correlation test). [Colour figure can be viewed at wileyonlinelibrary.com]

cell synapses is a major factor in determining the amplitude of simple spike modulation, and this strength is controlled by the timing of climbing fibre activity: the presence of climbing fibre activity during parallel fibre input leads to long-term depression, while its absence induces long-term potentiation (Coesmans et al., 2004; Gutierrez-Castellanos et al., 2017; Ohtsuki et al., 2009; Romano et al., 2018). Since molecular layer interneurons can suppress the complex spike-induced $Ca^{2+}$ influx, they may modulate the impact of complex spikes on parallel fibre plasticity (Rowan et al., 2018). The effect of parallel fibre plasticity on cerebellar output is reinforced by synergistic plasticity at other synapses in the cerebellum (D'Angelo et al., 2016; De Zeeuw, 2021; Gao et al., 2012; Geminiani et al., 2024). Thus, the interaction between the three main inputs to Purkinje cells – the climbing fibres, parallel fibres and molecular layer interneurons – regulates the depth of simple spike modulation, and thereby, cerebellar learning.

One might expect that, if a Purkinje cell is involved in any given orofacial task, it should engage with changes of both complex spike and simple spike firing. However, we observed that of the 60 Purkinje cells that showed any type of firing rate modulation during rhythmic licking, merely 32% revealed both complex spike and simple spike modulation. Moreover, of the Purkinje cells that engaged in a non-canonical way, 65% displayed only simple spike modulation and 3% only complex spike modulation (Fig. 1*D*). Thus, during spontaneous, rhythmic licking behaviour it was more common to find a relevance for simple spike modulation than that for complex spikes, which is similar for other spontaneous, rhythmic behaviours, such as respiration (Romano et al., 2020). In contrast, when unexpected events occur, such as the sudden movement of the lick port, the number of complex spikes increases, which in turn has consequences for the simple spike firing rate and thereby for the behaviour.

## The consequences of complex spike firing

To explain why many Purkinje cells encode spontaneous rhythmic licking with virtually only simple spikes, whereas they may show prominent modulations of not only simple spikes but also of complex spikes when the animals adjust the licking direction to a moving lick port, it is important to further examine the existing theories of complex spike function during motor learning. Probably the most popular of these theories relates to the internal models of body dynamics that are used by the cerebellum to create predictions of sensory feedback of planned actions (McNamee & Wolpert, 2019; Smith & Shadmehr, 2005; Therrien & Bastian, 2015). During learning climbing fibre activity can signal a mismatch between the predicted and perceived sensory feedback, akin to an error signal, and thereby induce long-term depression of parallel fibre synapses. In this way, the encoding of sensory prediction errors by climbing fibre activity can lead to recalibration of the internal model to ultimately improve behavioural performance (Spaeth & Isope, 2023; Streng et al., 2018; Tseng et al., 2007; Yang & Lisberger, 2014). When interpreting our observations that complex spikes were triggered by the sudden unexpected movement of the lick port and that this increase in complex spike firing was followed by an adaptation of the simple spikes and tongue trajectories during the subsequent licks, it is parsimonious to explain our data in the framework of this theory. Indeed, during the current adaptation process, during which the tongue movements were adjusted to the target, we also often observed the typical reciprocal occurrence of complex spikes and simple spikes (Fig. 7*F*), as previously observed in other behaviours (Graf et al., 1988; Romano et al., 2020). This highlights the dominant suppressing effects of complex spike firing (Badura et al., 2013; De Zeeuw et al., 2011; Tang et al., 2017; Zhou et al., 2014). Moreover, when the tongue movements were adjusted to the newly positioned

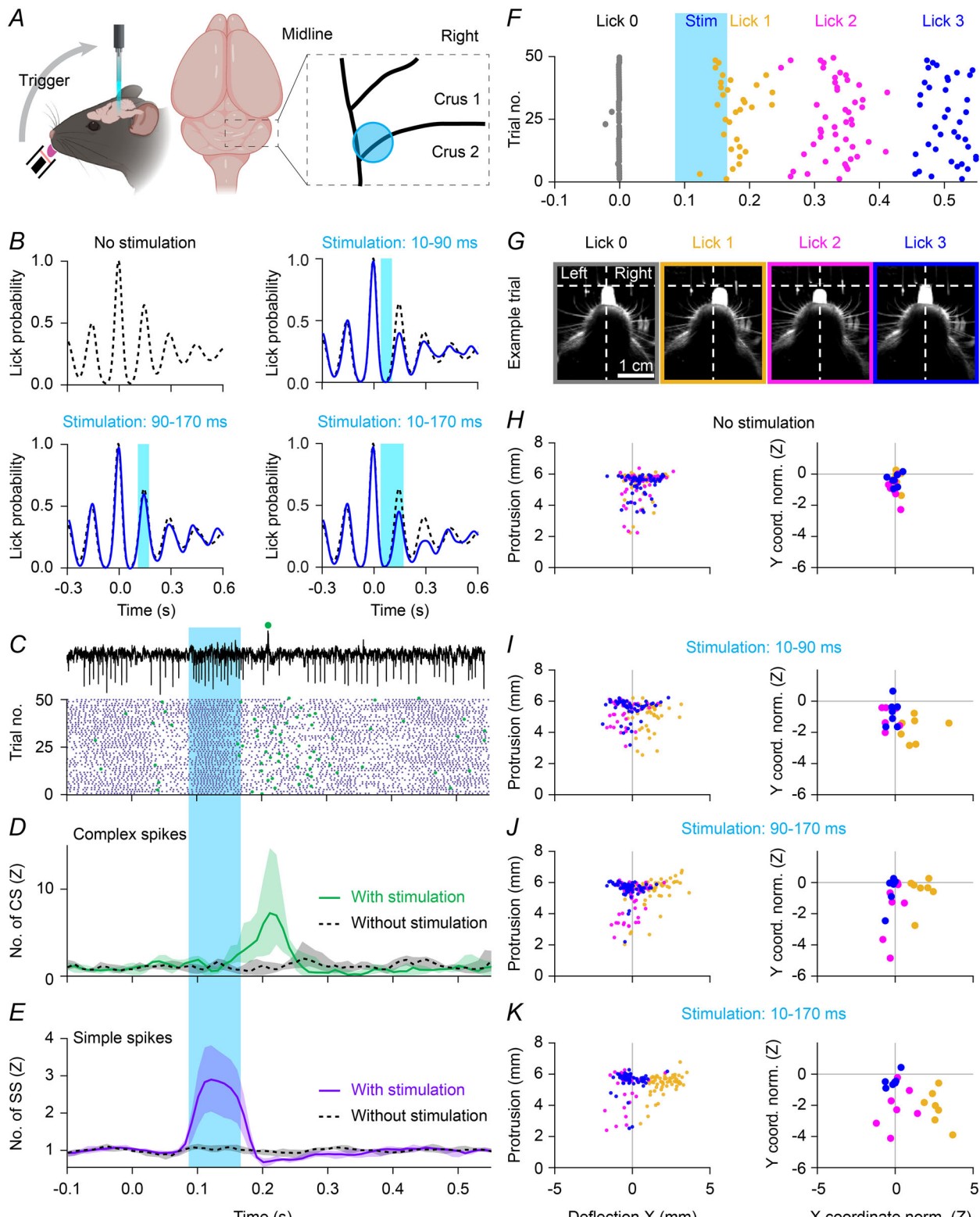

**Figure 8. Optogenetic stimulation affects the execution of licking**
*A*, optogenetic stimulation was performed by shining blue light (470 nm) on Purkinje cells expressing ChR2 under control of the *Pcp2* promotor. We used an optical fibre placed at the border of crus 1 and crus 2 close to the vermis on the right-hand side of the brain. Optogenetic stimuli were triggered by randomly selected licks, and we used three different light pulses: starting either at 10 or 90 ms after lick detection and lasting 80 ms, or a

longer pulse starting 10 ms after lick and lasting 160 ms. *B*, for an example mouse and for the three different optogenetic stimulation conditions, the distribution of licks that efficiently crossed the lick-port sensor (blue) *vs.* those in the absence of optogenetic stimulation (black). *C*, example of an extracellular recording of a Purkinje cell during optogenetic simulation and a raster plot of complex spikes (green) and simple spikes (purple) distribution around optogenetic stimulation. Medians with shades representing IQR. *D*, complex spike (CS) peri-stimulus time histograms for trials with and without optogenetic stimulation (*n* = 10 Purkinje cells in 7 mice). Medians with shades representing IQR. *E*, as *D*, but for the simple spikes (SS). *F*, timing of occurrence of maximal tongue protrusions around optogenetic stimulations for an example mouse obtained after video tracking, for stimulation 90–170 ms after randomly selected licks (lick 0, grey). *G*, example frames belonging to four consecutive licks of a trial during optogenetic stimulation of the right hemisphere. Notice that the first lick after stimulation is bent to the right, the second lick is shorter, the third lick is comparable to the one before stimulation. *H*, left: for an example mouse, coordinates of licks 1, 2 and 3 (okra, magenta and blue, respectively) following randomly selected control licks not followed by optogenetic stimulation. Right: each datapoint is now representing the mean value of individual mice (*n* = 7 mice). norm., normalized. *I*, same as in *F*, but for optogenetic stimulations 10–90 ms after detection of lick zero (right, *x*-coordinate lick 1 *vs.* lick 0: *P* = 0.0168, paired *t* test with Bonferroni correction; *n* = 7 mice). *J*, same as in *F*, but for optogenetic stimulations 90–170 ms after detection of lick zero (right, *x*-coordinate lick 1 *vs.* lick 0: *P* < 0.0001). *K*, same as in *F*, but for optogenetic stimulations 10 to 170 ms after detection of lick zero (right, *x*-coordinate lick 1 *vs.* lick 0: *P* < 0.0001). [Colour figure can be viewed at wileyonlinelibrary.com]

target, the complex spikes decreased as the error signals were no longer necessary.

In more complex learning paradigms, when there are early cues that announce upcoming events, complex spikes can also act as a reinforcement signal (Bina et al., 2021; Ohmae & Medina, 2015; Ten Brinke et al., 2015). These context-dependent complex spikes can serve for example a role in associative learning, such as reflex conditioning and reward expectation (Heffley & Hull, 2019; Kostadinov et al., 2019; Larry et al., 2019; Medina et al., 2002; Ohmae & Medina, 2015; Romano et al., 2018, 2020; Rowan et al., 2018; Silva et al., 2024; Ten Brinke et al., 2015), where they signal the perceived salience of the internal and/or external inputs (Bina et al., 2021; Larry et al., 2019; Ten Brinke et al., 2015). Different from the complex spikes encoding error signals highlighted above, these complex spikes emerge after preceding cues have been sufficiently learned to be recognized and they occur exactly at the moment when consequential motor actions have to be taken, possibly facilitating the initiation of a new motor pattern (Bina et al., 2021).

Similarly, increased complex spike firing can also reflect a change in behavioural state (Hoogland et al., 2015; Kitazawa et al., 1998; Markanday et al., 2021; Streng et al., 2017; Wagner et al., 2021). This relation has been proposed to facilitate the switch to an internal model of the new behavioural state (Streng et al., 2022). Given that changes in complex spike activity occurred not only at the start, but also at the end of a licking bout, our data are in line with this possibility. Indeed, when being exposed in the set-up, the mice may have sufficient cues to understand that they are entering a new behavioural state, which may require specific spatiotemporal patterns of complex spike activity (Herzfeld et al., 2015; Hoogland et al., 2015; Markanday et al., 2021; Welsh et al., 1995).

How the various complex spike modulations in the cerebellar Purkinje cells during licking are ultimately read out in the cerebellar nuclei remains to be elucidated (Lu et al., 2013). As shown for eyeblink conditioning, presumptively synchronous complex spike activity of multiple Purkinje cells can elicit consecutive pauses and rebounds of activity in the cerebellar nuclei, which appear rather effective in facilitating a conditioned response (Broersen et al., 2023; Ten Brinke et al., 2017). This mechanism, which may set the complex spikes apart from comparable bursts of simple spike activity, could in principle play a role when the tongue has to be repositioned with respect to the newly placed target. In short, complex spikes may serve multiple functions related to changes during licking behaviour, but they probably play their most important role during adjustments of the tongue movements.

## Multitasking in cerebellum

Licking is strongly related to lip and jaw movements, swallowing, and respiration, but also to head movements and postural control. For efficient orofacial control, it is essential that all these behaviours are coordinated. This coordination is largely done by the cerebellum, as in other types of behaviour (Diener et al., 1992; Ilg et al., 2007; Jaarsma et al., 2024; Thach et al., 1992; Yoshida et al., 2022). One of the essential features of the cyto-architectonics of the olivocerebellar system is that the matrix of the sagittal climbing fibre system and the orthogonally organized mossy fibre–parallel fibre system in the cerebellar cortex allows for acquisition of a multitude of associations between myriad sensory inputs and many different motor outputs. Moreover, the system is highly dynamic with specific associative processes only played out over time under particular circumstances (Bina et al., 2021; Ohmae & Medina, 2015; Ten Brinke et al., 2015). Thus, while specific combinations of complex spike and simple spike modulations are required for particular forms of cerebellar learning, it is likely that the related synaptic weight changes affect multiple, related forms

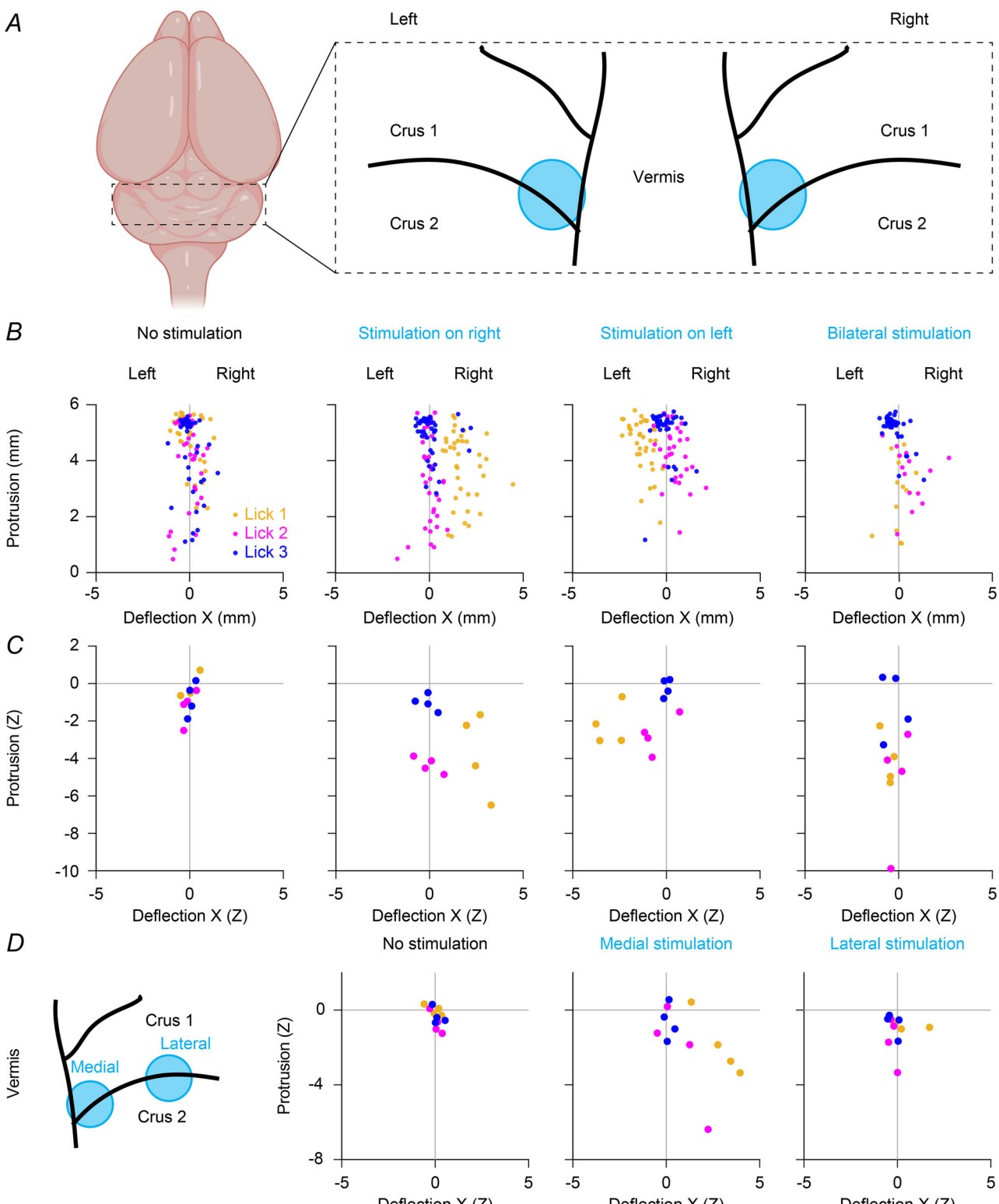

**Figure 9. Stimulation of the medial hemispheres induces an ipsilateral bias in tongue movements**
*A*, two optic fibres were placed on the right and on the left side of vermis, between lobules crus 1 and crus 2, and were activated either alone or simultaneously during licking. *B*, for an example mouse, coordinates of licks 1, 2 and 3 (okra, magenta and blue, respectively) in trials without optogenetic stimulation, with stimulations of the right hemisphere, with stimulations of the left one, and with simultaneous stimulations of both sides. *C*, same as in *B*, but each datapoint now represents the mean per mouse (*x*-coordinate, lick 1 stimulus on the right. *vs.*

lick 1 stimulus on the left, $P < 0.0001$, paired $t$ test; $n$ = 4 mice). *D*, for a different set of four mice, one fibre was placed on the right side as for previous experiments, and a second fibre was placed more laterally (left). Lateral stimulations were less effective (*x*-coordinate, lick 1 medial stimulus *vs*. lick 1 lateral stimulus, $P = 0.0220$; $n$ = 4 mice). [Colour figure can be viewed at wileyonlinelibrary.com]

of behaviours, and that these effects change over time, dependent on the context. This notion is in line with the multitude of behavioural parameters that are encoded in the crus regions of the cerebellar cortex (Bina et al., 2021; Bosman et al., 2010; Bryant et al., 2010; Cao et al., 2012; Gaffield et al., 2022; Heffley & Hull, 2019; Hoang et al., 2023; King et al., 2019; Kitazawa et al., 1998; Kostadinov et al., 2019; Lackey et al., 2024; Lindeman et al., 2021; Romano et al., 2020, 2022; Shambes et al., 1978; Streng et al., 2018; Wagner et al., 2021).

### Cerebellum and symmetry of movements

The coordinating function of the cerebellum in motor control is particularly relevant for movement symmetry. Healthy humans walking on a split-belt treadmill can readily adapt the stepping pattern of their right and left leg separately to specific speeds. Patients with cerebellar damage have trouble in generating predictive compensation mechanisms (Hoogkamer et al., 2015; Morton & Bastian, 2006). A similar inability to create asymmetric gaiting patterns was observed in cerebellar mutant mice (Darmohray et al., 2019). Accordingly, cerebellar activity has been shown to be required to coordinate the level of symmetry of whisker movements (Romano et al., 2022; Towal & Hartmann, 2006). Since we show in the present study that changes in complex spike firing can also be correlated to the level of symmetry of tongue movements, we propose that the cerebellum is crucial for controlling various types of (a)symmetric movements. The mechanisms implicated may depend on the specific structure of the related motor plant as well as on the synaptic polarity of the intermediate neuronal hubs downstream of the upbound and downbound modules involved (De Zeeuw, 2021). Given that we found a strong impact of increases in simple spike firing on determining the angle of individual licks within a licking bout, it may be particularly the upbound modules that are critical for controlling tongue movements.

### Tongue movements in different species

Mobile tongues probably evolved as an adaptation to feeding on land, and they can be found in virtually all tetrapods, with amphibian tadpoles that are aquatic as the major exception (Iwasaki, 2002). Tetrapod tongues, like trunks and tentacles, lack a skeleton and are essentially muscular hydrostats that can deform without volume change (Gilbert et al., 2007; Kier & Smith, 1985; Olson

et al., 2021). The latter implies that, as tongues cannot compress, muscle contraction at one place leads to shape changes at other places of the tongue, allowing complex movements to be made (Gilbert et al., 2007; Kier & Smith, 1985; Olson et al., 2021; Ross et al., 2024).

Tetrapod tongues vary widely in their anatomy, but most mammalian tongues share the same general structure and musculature (Sonntag, 1925). However, mammals that use their tongue primarily for extra-oral feeding, i.e. monotremes, marsupials and pangolins, possess long tongues that differ in shape and flexibility from those of the other mammals (Doran & Baggett, 1971; Hiiemae & Palmer, 2003; Kier & Smith, 1985; Sonntag, 1924b, 1925). The shape, structure and function of tongues are adapted to the environment, behaviour and common food sources of each species, so that even related species can differ to some extent in their tongue anatomy (Doyle et al., 2023; Erdoğan & Iwasaki, 2014; Sonntag, 1924a, 1925).

Most mammalian tongues thank for their motility a combination of four extrinsic and four intrinsic muscles that are often paired (Iwasaki et al., 2019; Sanders & Mu, 2013). Extrinsic muscles connect the tongue with surrounding bones, and intrinsic muscles are located entirely within the tongue. One could consider the longitudinal fibres of the genioglossus muscle as the driving force of tongue protrusion, and the hyoglossus muscle as the main actor of tongue retraction, but tongue movements are typically the result of simultaneous activity of multiple tongue muscles (Bennett, 1937; Oliven et al., 2025). The initial conclusion that left- or rightward bending of the tongue results solely from unilateral activation of the genioglossus muscle (Braus, 1924) is therefore likely an oversimplification. In reality, deflections of the tongue are caused by the asymmetric activation of several tongue muscles (Abd-El-Malek, 1938; Bennett, 1937; Lowe, 1980).

Most terrestrial vertebrates use either suction or licking to acquire water. When licking, the tongue makes rhythmic movements to transport water into the mouth, while tongue movements are restricted to intra-oral transport of liquids when suckling (Reis et al., 2010; Thexton et al., 1998; Weijnen, 1998). Suction is the primary method of milk intake in all infant mammals, as well as of water intake in adult mammals with complete cheeks, such as pigs and humans. Mammals with incomplete cheeks, such as cats, dogs, and mice, transition from suckling to licking after weaning (Mayerl et al., 2021; Thexton et al., 1998).

In addition to its role in ingestion, mammalian tongues are also important for respiration (Krohn et al., 2023;

Oliven et al., 2025) and gustation (Davydova et al., 2017; Doyle et al., 2023; Kobayashi et al., 1989; Kumari & Mistretta, 2023; Münch, 1896; Sewards, 2004; Spence, 2022). In some species, tongues may also serve other functions, like temperature control in dogs (Kindermann & Pleschka, 1973) and speech in humans (Ekström & Edlund, 2023; Lieberman, 2007). To facilitate speech production human tongues are unusually rich in slow twitch muscle fibres and loose connective tissue (Sanders et al., 2013). In addition, the human tongue root is located more downwards into the throat, unlike in any other mammal, which is also likely an adaptation to speech production (Ekström & Edlund, 2023; Lieberman, 2007). Notably, non-human mammals do not actively modulate sounds with their tongues (Ekström & Edlund, 2023; Fitch, 2000).

Thus, tetrapod tongues share the same basic design as muscular hydrostats and most mammals have similar muscularization of their tongues. Nevertheless, anatomical details can vary even between closely related species, such as the different Muridae (Sonntag, 1924a), and behaviour can vary even more widely. It will, therefore, be interesting to compare the cerebellar coordination of tongue movements in different species.

### Innervation of the tongue muscles

Most tongue muscles are controlled by motor neurons in the hypoglossal nucleus. Only the (extrinsic) palatoglossus muscle is innervated from the vagus nerve. Accordingly, the hypoglossal motor neurons produce rhythmic activity in tune with tongue movements (Wiesenfeld et al., 1977). A plausible origin of this rhythmic activity is a group of the homeobox gene *Phox2b*-positive pre-motor neurons in the intermediate reticular formation (Dempsey et al., 2021), which is the brain area supplying most of the afferent fibres to the hypoglossal nucleus (Guo et al., 2020). In itself, the brainstem circuit is likely to be sufficient for the generation of reflexive rhythmic tongue movements, but it does require input from the cerebral cortex for volitional tongue movements (Bignall & Schramm, 1974; Bollu et al., 2021; Grill & Norgren, 1978; Kleinfeld et al., 2023) and, as we show here, the cerebellum for adjustments of phase and amplitude.

### Projections from cerebellum to hypoglossal nucleus

Purkinje cells project predominantly to the cerebellar nuclei, and, consequently, cerebellar nucleus neurons are involved in licking as well (Lu et al., 2013). From the cerebellar nuclei, there are multiple pathways along which the cerebellum can affect the execution of tongue movements. First of all, there is a direct but sparse projection to the hypoglossal nucleus (Guo et al., 2020; Judd et al., 2021; Novello et al., 2024). Indirect projections between the cerebellum and the hypoglossal nucleus may be numerically and functionally more relevant, as in other motor systems (Novello et al., 2024). Importantly, the cerebellum provides many of the non-brainstem inputs to the *Phox2b*-positive pre-motor neurons in the intermediate reticular formation (Dempsey et al., 2021), making this cluster of neurons, which is considered a central pattern generator for tongue movements, a likely intermediate for cerebellar output. Given that the cerebellum broadly projects to brainstem regions (Lu et al., 2013; Novello et al., 2024; Teune et al., 2000), and that many, if not most, of these project to the hypoglossal nucleus (Guo et al., 2020), also other brainstem-mediated indirect pathways may serve to convey cerebellar output to the hypoglossal nucleus. Finally, the cerebellum could also modulate tongue movements indirectly via the thalamus and tongue motor cortex (Aoki et al., 2019).

### Clinical relevance

The inability to control the tongue properly can lead to dysarthria, dysphagia and respiratory problems including obstructive sleep apnoea, which, in turn, can have a serious impact on the quality of life, and, via aspiration pneumonia, even be lethal (Daniels et al., 1999; Duan et al., 2020; Krohn et al., 2023; Oliven et al., 2025; Takizawa et al., 2016). There are multiple possible causes for oral movement disorders, of which impairment of the basal ganglia, as in Parkinson's and Huntington's diseases, is the best characterized (Leopold & Kagel, 1996; Minagi et al., 2018; Reilmann et al., 2010). Neurodegenerative diseases and stroke are among the diverse neurological conditions that can impair tongue function, and thereby cause problems with speech and ingestion (Takizawa et al., 2016; Tao et al., 2012). Notably, both anterior and posterior circulation infarcts can affect tongue motility (Tao et al., 2012). The impact of anterior circulation infarcts can probably largely be explained by the impact of cerebral motor areas, especially the facial area of the primary motor cortex, which projects directly to the hypoglossal nucleus in humans and other primates (Kuypers, 1958a,b; Jürgens & Alipour, 2002; Morecraft et al., 2014). In addition, there are also indirect projections from the motor cortices to the hypoglossal nucleus, possibly via the *Phox2b*-positive neurons of the intermediate reticular formation (Dempsey et al., 2021; Jürgens & Alipour, 2002). While the direct cortico-hypoglossal projections seem to be limited to primates, the indirect projections are also abundant in other mammals (Dempsey et al., 2021; Jürgens & Alipour, 2002).

Damage to the cerebellum could explain, at least in part, the impact of the posterior circulation infarcts on

the ability to control the tongue. Accordingly, problems with speech and swallowing are prevalent in patients with diverse forms of cerebellar ataxia (Giardina et al., 2020; Ikeda et al., 2012; Keage et al., 2017; Markovic et al., 2016; Rezende Filho et al., 2019; Tan et al., 2000; Ushe & Perlmutter, 2012; Woo et al., 2019). Functional brain imaging data are in line with the activation of the cerebellum during various tongue-related behaviours (Boillat et al., 2020; Corfield et al., 1999; Dimitrova et al., 2006; Grabski et al., 2012; Groenendijk et al., 2020; Ogura et al., 2012; Sörös et al., 2020). Likewise, cerebellar dysfunction can also be associated with hyperkinetic tongue movements (Salari et al., 2023). Tongue fasciculations, tremor and dystonia may occur in different forms of cerebellar ataxia, but predominantly in individuals that have also extracerebellar brain damage (Izumi et al., 2013; Salari et al., 2023). Thus, clinical observations imply the cerebellum in control of tongue movements, but not always as the primary cause of deficits thereof. The latter is in line with our observations. Similar to other motor behaviours (De Zeeuw et al., 2011), we failed to produce evidence that the cerebellum can initiate tongue movements, but we did find that the firing pattern of Purkinje cells can reflect specific aspects of tongue movement, including the timing, amplitude and angle of protrusion. Our data on both the timing of Purkinje cell activity and the impact of optogenetic stimulation suggest that degeneration of Purkinje cells will affect execution and adaption of tongue movements.

## Conclusions

Tongue movements can be complex, and precise control of the tongue is required for such important processes as ingestion, respiration and speech. Reflexive tongue movements, triggered by sensory stimulation, can be coordinated solely by the brainstem and rely heavily on a central pattern generator in the intermediate reticular formation. Volitional control of the tongue requires the primary motor cortex, probably in conjunction with accessory motor areas of the cerebral cortex. Instead, modulations of simple spikes and complex spikes of Purkinje cells in the cerebellum are essential for execution of fine motor control and adaptation to the behavioural context, respectively. Disruption of any of these brain regions leads to impairments in tongue movement control that can eventually be lethal.

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

## Additional information

### Data availability statement

Data associated with this study have been stored in a repository (https://data.mendeley.com/datasets/6n27597vdf/2).

### Competing interests

The authors declare they have no competing interests.

## Author contributions

Conception or design of the work: L.B., C.C., S.Y., L.W.J.B. and C.I.D.Z. Acquisition and analysis of data: L.B., C.C., S.Y., X.W. and L.W.J.B. Interpretation of data: L.B., C.C., S.Y., L.W.J.B. and C.I.D.Z. Drafting and revising article: L.B., S.Y., X.W., L.W.J.B. and C.I.D.Z. All authors have approved the final version of the manuscript and agree to be accountable for all aspects of the work in ensuring that questions related to the accuracy or integrity of any part of the work are appropriately investigated and resolved. All persons designated as authors qualify for authorship, and all those who qualify for authorship are listed above.

## Funding

This study was supported by Health-Holland to promote public-private partnerships (TKI-LSH EMCLSH21017: L.W.J.B.). C.I.D.Z. received financial funding from the European Union's Horizon 2020 research and innovation program under the Marie Skłodowska-Curie grant agreement (#722098), Medical NeuroDelta Programme, Topsector Life Sciences & Health (Innovative Neurotechnology for Society or INTENSE), Albinism Vriendenfonds Netherlands Institute for Neuroscience, and European Research Council – Advanced Grant (#294775), NWO-Gravitation Program (DBI2).

## Acknowledgements

The authors thank Stéphanie Dijkhuizen and Nathalie van Wingerden for excellent technical support.

## Keywords

cerebellum, complex spikes, licking, motor control, Purkinje cells, simple spikes, tongue

## Supporting information

Additional supporting information can be found online in the Supporting Information section at the end of the HTML view of the article. Supporting information files available:

**Peer Review History**

