## [Peer Review History · The Journal of Physiology]

Cerebellar control of targeted tongue movements

Lorenzo Bina, Camilla Ciapponi, Si-yang Yu, Xiang Wang, Laurens W. Bosman, and Chris De Zeeuw
DOI: 10.1113/JP287732

Corresponding author(s): Laurens Bosman (l.bosman@erasmusmc.nl)

Review Timeline:

Submission Date:	25-Sep-2024
Editorial Decision:	11-Nov-2024
Revision Received:	20-Dec-2024
Accepted:	10-Jan-2025

Senior Editor: Harold Schultz

Reviewing Editor: Jing-Ning Zhu

Transaction Report:

Dear Dr Bosman,

Re: JP-RP-2024-287732 "Cerebellar control of targeted tongue movements" by Lorenzo Bina, Camilla Ciapponi, Si-yang Yu, Xiang Wang, Laurens W. Bosman, and Chris De Zeeuw

Thank you for submitting your manuscript to The Journal of Physiology. It has been assessed by a Reviewing Editor and by 2 expert referees and we are pleased to tell you that it is potentially acceptable for publication following satisfactory major revision.

REVISION CHECKLIST:

Please upload two versions of your manuscript text: one with all relevant changes highlighted and one clean version with no

changes tracked. The manuscript file should include all tables and figure legends, but each figure/graph should be uploaded as separate, high-resolution files.

We look forward to receiving your revised submission.

Yours sincerely,

Harold Schultz
Senior Editor
The Journal of Physiology

REQUIRED ITEMS

- Author photo and profile. First or joint first authors are asked to provide a short biography (no more than 100 words for one author or 150 words in total for joint first authors) and a portrait photograph. These should be uploaded and clearly labelled together in a Word document with the revised version of the manuscript. See Information for Authors for further details.

- You must start the Methods section with a paragraph headed Ethical approval (https://jp.msubmit.net/cgi-bin/main.plex?form_type=display_requirements#methods).

Research must comply with The Journal's policies regarding animal experiments (<https://physoc.onlinelibrary.wiley.com/hub/animal-experiments>) and adherence to these policies must be stated in the manuscript.

Authors should confirm in their Methods section that their experiments were carried out according to the guidelines laid down by their institution's animal welfare committee, including an ethics approval reference number. The Methods section must contain a statement about access to food, water and housing, details of the anaesthetic regime: anaesthetic used, dose and route of administration, and method of killing the experimental animals.

- The Journal of Physiology funds authors of provisionally accepted papers to use the premium BioRender site to create high resolution schematic figures. Follow this link and enter your details and the manuscript number to create and download figures. Upload these as the figure files for your revised submission. If you choose not to take up this offer, we require figures to be of similar quality and resolution. If you are opting out of this service to authors, state this in the Comments section on the Detailed Information page of the submission form. The link provided should only be used for the purposes of this submission. Authors will be charged for figures created on this premium BioRender account if they are not related to this manuscript submission.

- Please upload separate high-quality figure files via the submission form.

- Please ensure that the Article File you upload is a Word file.

- Papers must comply with the Statistics Policy: https://jp.msubmit.net/cgi-bin/main.plex?form_type=display_requirements#statistics.

In summary:

- If n {less than or equal to} 30, all data points must be plotted in the figure in a way that reveals their range and distribution. A bar graph with data points overlaid, a box and whisker plot or a violin plot (preferably with data points included) are acceptable formats.

- If $n > 30$, then the entire raw dataset must be made available either as supporting information, or hosted on a not-for-profit repository, e.g. FigShare, with access details provided in the manuscript.

- 'n' clearly defined (e.g. x cells from y slices in z animals) in the Methods. Authors should be mindful of pseudoreplication.

- All relevant 'n' values must be clearly stated in the main text, figures and tables.

- The most appropriate summary statistic (e.g. mean or median and standard deviation) must be used. Standard Error of the Mean (SEM) alone is not permitted.

- Exact p values must be stated. Authors must not use 'greater than' or 'less than'. Exact p values must be stated to three significant figures even when 'no statistical significance' is claimed.

- Please include an Abstract Figure file, as well as the Figure Legend text within the main article file. The Abstract Figure is a piece of artwork designed to give readers an immediate understanding of the research and should summarise the main conclusions. If possible, the image should be easily 'readable' from left to right or top to bottom. It should show the physiological relevance of the manuscript so readers can assess the importance and content of its findings. Abstract Figures should not merely recapitulate other figures in the manuscript. Please try to keep the diagram as simple as possible and without superfluous information that may distract from the main conclusion(s). Abstract Figures must be provided by authors no later than the revised manuscript stage and should be uploaded as a separate file during online submission labelled as File Type 'Abstract Figure'. Please also ensure that you include the figure legend in the main article file. All Abstract Figures should be created using BioRender. Authors should use The Journal's premium BioRender account to export high-resolution images. Details on how to use and access the premium account are included as part of this email.

- Please include a full title page as part of your main article (Word) file, which should contain the following: title, authors, affiliations, corresponding author name and contact details, keywords, and running title.

Reviewing Editor's comments:

This interesting manuscript explores some encoding characteristics of complex spike and simple spike discharges of Purkinje cells for tongue motor coordination. Please revise the manuscript according to the suggestions of two reviewers, including a discussion of the role of complex spikes in error signaling and state changes, the multi-tasking of individual Purkinje cells, and the finding of the computational model. Especially, why the complex spikes increase significantly before licking but do not predict licking by the computational model needs to be discussed in detail. Furthermore, reconstruction of the locations of recorded Purkinje cells is encouraged.

Senior Editor:

Comments to ensure the paper complies with the Statistics Policy:

Actual p-values need to be shown in figures. Please do not use symbols.

Comments to the Author:

Thank you for submission of your research article to the Journal of Physiology for consideration. The article has been reviewed by experts in the field and found to require a major revision to address all of the concerns raised before further assessment can be given. Please address all comments from the external referees and reviewing editor as well as addressing the list of requirements or publication in the journal as described in this letter.

In addition, the manuscript must comply with the Journal's policies on Rigour and Reproducibility as described in the document provide with the link below.

<https://physoc.onlinelibrary.wiley.com/pb-assets/hub-assets/physoc/documents/TJP-Rigour-and-Reproducibility-Requirements-1724673661727.pdf>

Some specific concerns (but not exclusive) are detailed below:

Surgical Procedures: Start the section with describing the method of anesthesia, and maintenance of surgical plane including how the plane was assessed.

Electrophysiology to record Purkinje cell activity requires much more detail of the surgical method of anesthesia, maintenance of surgical plane including how the plane was assessed, and details of the surgical approach, how the electrodes were fixed in place, etc. Post surgical procedures to minimize pain and distress need to be described and assured.

Optogenetic stimulation: The same concerns as above for electrophysiology need to be addressed.

The end points of experiments on animals need to be clearly stated. The method of termination of the animal and method of confirmation of death need to be provided.

Referee #1:

This is a highly interesting paper from the Bosman and De Zeeuw groups at Erasmus MC. Tongue movements make a very interesting case for the study of cerebellar movement control and also for the study of a cerebellar involvement in screening the sensory environment. The present study focuses on Purkinje cell firing patterns, and a distinction of complex vs simple spike firing, during lick movements as well as during adaptation to a new position of the lick port. A main finding of the study is that simple spike firing is well correlated with tongue movement and projection, while complex firing is less correlated. In

contrast, complex spikes are reliably fired when the lick port is quickly and surprisingly moved; this is in line with the role of CF activity in error encoding.

There is a lot to take away from this study for many cerebellar researchers (and beyond), and for every individual it will vary what they find most interesting. The study provides a lot of data building on previous work mostly from the lab of Detlef Heck.

What interests this reviewer is the multi-tasking that individual Purkinje cells can do. So let's start with some aspects related to that.

a) Fig. 1: The authors state that they are surprised that only 23% of Purkinje cells in their 'field of view' show modulated ss and cs responses. I find that number actually quite high, particularly in areas VI, VII, Crus1, II where Purkinje cells are known to be engaged in many tasks, including responding to sensory stimuli of various modalities. If the authors had studied cerebellar involvement in the movement of hands, arms and legs, they would still see similar percentages of Purkinje cells being engaged, would they not? If that is so, the question becomes how this multitasking can actually work. Or doesn't it and we are looking at the wrong thing? Is there a chance that some of the differences in firing rates are actually related to sensory experience rather than motor output? (Every time the tongue ejects there is motor action and new sensory experience.) Perhaps the authors can discuss multitasking a bit more in their paper.

b) Fig. 7: This is another interesting finding: upon fast movement of the lick port, complex spikes are fired in 42% of Purkinje cells. That is strong evidence for the classical 'error' encoding idea (unless PCs encode the stimulus movement itself). Maybe the authors should discuss error coding a bit more, particularly since this classical idea has lost traction after new complex spike functions were discovered, e.g. in the context of reward encoding.

c) Back to multi-tasking. The authors state that in individual Purkinje cells, complex spike firing may occur when the target moves and simple spike firing encodes the tongue movement itself. That is highly interesting. First, why does it make sense to encode these things in the same cells? Second, would not a complex spike be followed by a pause, which would prevent proper simple spike encoding for 50-200ms? Third, thinking of downstream neurons: is a complex spike really that different from a ss burst? If so, this becomes a circuit effect (several Purkinje cells and their convergence effects). Can the authors help to make more sense here and add this to the discussion so that everybody can benefit?

d) For the relevance to humans / clinical applications: Are tongue movements / sensations different in primates than in rodents (for example)? Same number of muscles? Taste receptors? Is there something to make the argument that tongue movement is particularly complex in humans, because of its role in language?

Referee #2:

This manuscript by the DeZeeuw lab examines targeted tongue movements in the mouse and its relationship to Purkinje cell discharge. The basic observations are that both the simple spike discharge and complex spike discharge are modulated in relationship to tongue movements but with different modulation characteristics, as one would expect as Purkinje cells are modulated with virtually all movements. The study also uses optogenetic manipulations to test how activation of Purkinje cells modulates tongue movements. The work is interesting as the first detailed examination of this question. However, it's not quite clear what the overall purpose of the study was beyond an exploration of the topic and wish that there was a more specific hypothesis driving the work. Still the study has important value, as providing fundamental characteristics of Purkinje cell firing in relation to tongue movements.

There are several major questions and comments that need to be addressed by the authors. The computational model does not seem to add very much to the findings or understanding of the role of either complex spike or simple spike firing in tongue movements. The relationship revealed by the model seemed to be somewhat weak. Similarly, it is hard to understand the meaning of the decoding contribution of the Purkinje cell parameters. For example, the statement on line 337 says that 69% of Purkinje cells had at least some relation to licking behavior, with the word "some" not very precise. Also, given the modulation observed in the complex spikes, both at the beginning and end of the lick in Figure 2, it was surprising that the model found that complex spike firing was a poor predictor of licking. Also, what does it mean that in Figure 3A that there is no simple spike CV2 bias between licking and no licking? Or that there is no bias for simple spike frequency in Figure 2B.

Also, the model was based on exceeding a SHARP threshold of 65. This needs to be explained better, including why that threshold was used, implications for statistical significance, and how to interpret.

Another concern is that the role of complex spikes in error detection or error signaling argued for in the Discussion, specifically as providing a sensory prediction error in the experiment in which the lick port was moved. For the experiments in which the click port was moved, the complex spike firing showed increases both to the right and the left but did not discriminate between the two directions. This lack of directionality suggests that the complex spikes are not error signaling as there is no information about the specifics of the movement but again possibly a change of state. Is this how the authors are interpreting this finding?

Instead, the findings suggest a role for the complex spikes in a change of state that was mentioned in the paper, which has also been suggested by Streng et al. *Journal of Neuroscience*, 2017 and *Cerebellum* in 2022. The change of state should be addressed given the present findings. Similarly, the fact that the complex spikes are show modulation related to the control of normal tongue movements without any obvious errors, also suggests that the complex spikes have a role in specifying movement parameters, as suggested by a number of recent papers. This issue should also be discussed in the manuscript.

Concerning the topography of the responses shown in Figure 4B and D, asking the question whether there is a specific distribution of the simple spike and complex spike responses depends on that the sampling of Purkinje cells was uniform across the region of the cerebellar cortex studied. Reviewers need to know the locations of the Purkinje cells recorded on those maps to interpret those findings. Also, as only a small number of Purkinje cells were recorded in each mouse this may make it harder to make such a spatial determination and statistical testing is needed.

Specific Comments:

In the first Paragraph of the results, lines 279 to 283, the comment is made that Pekingese cells can be defined as task related when they display both complex spike and simple spike modulation. It was unclear why the manuscript makes this point or how it fits into the overall results and discussion. Especially as they only find 19% of the Pekingese cells having both complex spikes and simple spike modulation in relation to the tongue movements.

Concerning that complex spike modulation in relation to tongue movements, there is an increase in complex spikes at the onset as shown in Figure 2A and B. The text then states in lines 309 and 310 that there's an increase in complex firing in 51% of Purkinje cells also at the end of the licking bout. However, in Figure 2C and D it appears the complex firing decreases, not increases. Please clarify. Also, for the plots in Figure 2B and D please label the time points that are actually significant modulation by the criteria used, as few of the points in D seem to ± 3 Zscores. Also, what exactly does the rectangular boxes in B&D signify? Also needed is when the licking starts and ends and the variability of those time points.

In Figure 4 and the associated analysis a coherence of greater than 0.5 was used as the threshold for a coherence. How was this threshold chosen and does it relate in any way to statistical significance? Also, in Figure 4F the claim is made that the simple and complex spikes are in antiphase. Was this based on a statistical test or is this a qualitative statement?

In Figure 8 the optogenetic stimulation shows modulation in both the tongue movement and the simple and complex firing. Did the optogenetic stimulation generate any other movements or twitches, particularly in the oral facial areas, given that the Purkinje cells were targeted widely. Stimulation 10-90 ms (I) is in-between licks, while 90-170 ms (J) is during lick execution and the effects are similar. This sort of confounding when it comes to control mechanisms, can the authors comment? Also the 10-90 ms stimulation reduces the next lick probability. Is this consistent with CS involvement in lick bouts?

END OF COMMENTS

Reviewing Editor's comments:

This interesting manuscript explores some encoding characteristics of complex spike and simple spike discharges of Purkinje cells for tongue motor coordination. Please revise the manuscript according to the suggestions of two reviewers, including a discussion of the role of complex spikes in error signaling and state changes, the multi-tasking of individual Purkinje cells, and the finding of the computational model. Especially, why the complex spikes increase significantly before licking but do not predict licking by the computational model needs to be discussed in detail. Furthermore, reconstruction of the locations of recorded Purkinje cells is encouraged.

Authors' response:

We would like to thank both Editors and both Reviewers for the attention given to our manuscript, for noting the interesting findings described, and for the constructive comments.

We have revised our manuscript according to the suggestions provided. In our revision, we have included the points highlighted here by the Reviewing Editor: a discussion of the roles of complex spikes, a discussion of the ability of Purkinje cells to engage in multiple tasks, a better integration of the model with the rest of the manuscript, and we have added the locations of the recorded Purkinje cells to Fig. 4.

In particular, we now explain the limitations of the model better, explaining the observed discrepancy between the role of complex spikes in behavioural state changes and the absence of predictive power of complex spike firing per 200 ms interval.

We hope that the revised version will be acceptable for publication in The Journal of Physiology.

Senior Editor:

Comments to ensure the paper complies with the Statistics Policy: Actual p-values need to be shown in figures. Please do not use symbols.

Comments to the Author:

Thank you for submission of your research article to the Journal of Physiology for consideration. The article has been reviewed by experts in the field and found to require a major revision to address all of the concerns raised before further assessment can be given. Please address all comments from the external referees and reviewing editor as well as addressing the list of requirements or publication in the journal as described in this letter.

In addition, the manuscript must comply with the Journal's policies on Rigour and Reproducibility as described in the document provide with the link below.

<https://physoc.onlinelibrary.wiley.com/pb-assets/hub-assets/physoc/documents/TJP-Rigour-and-Reproducibility-Requirements-1724673661727.pdf>

Authors' response:

We thank the Senior Editor for the attention given to our manuscript, for allowing us to submit a revised version, and for highlighting those aspects of our methods that need clarification. We have revised the text accordingly.

Some specific concerns (but not exclusive) are detailed below:

Surgical Procedures: Start the section with describing the method of anesthesia, and maintenance of surgical plane including how the plane was assessed.

Authors' response:

The description of the surgical procedure now starts with the details on anaesthesia, and is now much more detailed.

Electrophysiology to record Purkinje cell activity requires much more detail of the surgical method of anesthesia, maintenance of surgical plane including how the plane was assessed, and details of the surgical approach, how the electrodes were fixed in place, etc. Post surgical procedures to minimize pain and distress need to be described and assured.

Authors' response:

The description of electrophysiological recording procedures has been updated to address the issues raised by the Senior Editor. Note that we made acute recordings; the electrodes were not implanted or fixed. The text has been amended to clarify this.

Optogenetic stimulation: The same concerns as above for electrophysiology need to be addressed.

Authors' response:

To clarify the procedures, we have updated our text.

The end points of experiments on animals need to be clearly stated. The method of termination of the animal and method of confirmation of death need to be provided.

Authors' response:

The method of termination is now described in the text, along with the procedure to confirm death.

Referee #1:

This is a highly interesting paper from the Bosman and De Zeeuw groups at Erasmus MC. Tongue movements make a very interesting case for the study of cerebellar movement control and also for the study of a cerebellar involvement in screening the sensory environment. The present study focuses on Purkinje cell firing patterns, and a distinction of complex vs simple spike firing, during lick movements as well as during adaptation to a new position of the lick port. A main finding of the study is that simple spike firing is well correlated with tongue movement and projection, while complex firing is less correlated. In contrast, complex spikes are reliably fired when the lick port is quickly and surprisingly moved; this is in line with the role of CF activity in error encoding.

Authors' response:

We would like to thank Reviewer 1 for the positive and constructive comments.

There is a lot to take away from this study for many cerebellar researchers (and beyond), and for every individual it will vary what they find most interesting. The study provides a lot of data building on previous work mostly from the lab of Detlef Heck.

Authors' response:

Indeed, the work from Detlef Heck and his colleagues has been a strong inspiration and starting point for this study. We are happy to read that Reviewer 1 appreciates our results on cerebellar movement control and sensorimotor integration.

What interests this reviewer is the multi-tasking that individual Purkinje cells can do. So let's start with some aspects related to that.

a) Fig. 1: The authors state that they are surprised that only 23% of Purkinje cells in their 'field of view' show modulated ss and cs responses. I find that number actually quite high, particularly in areas VI, VII, CrusI, II where Purkinje cells are known to be engaged in many tasks, including responding to sensory stimuli of various modalities. If the authors had studied cerebellar involvement in the movement of hands, arms and legs, they would still see similar percentages of Purkinje cells being engaged, would they not? If that is so, the question becomes how this multitasking can actually work. Or doesn't it and we are looking at the wrong thing? Is there a chance that some of the differences in firing rates are actually related to sensory experience rather than motor output? (Every time the tongue ejects there is motor action and new sensory experience.) Perhaps the authors can discuss multitasking a bit more in their paper.

Authors' response:

It is indeed remarkable, maybe even astonishing, how many different aspects of animal behaviour are represented in crus 1 and surrounding areas of the cerebellar cortex. We fully agree with Reviewer 1 on both aspects mentioned in this comment: the involvement of this part of the cerebellum in multiple tasks as well as in the sensory and motor aspects of these tasks. Reviewer 2 (see comment no. 5) made a similar remark.

We have added a section on "Multitasking in the cerebellum" to the Discussion to place tongue movements in the context of related movements. In our opinion, a strong coordination between movements of the tongue, jaws, lips, pharynx and neck is required to optimize licking, as there needs to be a negative coordination between licking and respiration. As

such, it may be unsurprising that so many motor tasks reside in this part of the cerebellar cortex, often involving the same Purkinje cells for different sets of muscles.

In addition, the role of sensory feedback is now more emphasized in the Discussion, in particular in the new section on “The consequences of complex spike firing”. Here, we discuss the necessity of sensory feedback to optimize motor control by the cerebellum, and we try to make here also the connection to the topic raised in the next comment of Reviewer 1: error vs. reward encoding (please see also our response to that comment).

b) Fig. 7: This is another interesting finding: upon fast movement of the lick port, complex spikes are fired in 42% of Purkinje cells. That is strong evidence for the classical 'error' encoding idea (unless PCs encode the stimulus movement itself). Maybe the authors should discuss error coding a bit more, particularly since this classical idea has lost traction after new complex spike functions were discovered, e.g. in the context of reward encoding.

Authors' response:

Indeed, there is quite some discussion lately on the role of complex spike firing. Given the overwhelming evidence in favour of the “error theory”, we do not think that this idea has lost traction. What is becoming clearer, however, is that complex spikes also serve other functions than encoding solely sensory prediction errors. We have placed our findings in the context of older and recent work, naming four specific roles for complex spike firing in the new section “The consequences of complex spike firing” in the Discussion. We are aware that this is probably not a complete overview (which would also not fit the format of a research article) and are excited to learn about new roles for complex spikes from our and other research groups. See also comment no. 3 from Reviewer 2 and our response to that comment.

c) Back to multi-tasking. The authors state that in individual Purkinje cells, complex spike firing may occur when the target moves and simple spike firing encodes the tongue movement itself. That is highly interesting. First, why does it make sense to encode these things in the same cells? Second, would not a complex spike be followed by a pause, which would prevent proper simple spike encoding for 50-200ms? Third, thinking of downstream neurons: is a complex spike really that different from a ss burst? If so, this becomes a circuit effect (several Purkinje cells and their convergence effects). Can the authors help to make more sense here and add this to the discussion so that everybody can benefit?

Authors' response:

We would like to thank Reviewer 1 for bringing this up.

First, complex spikes have an important instructive role for parallel fibre plasticity, and it makes, therefore, sense to have a combination of sensory feedback (complex spikes) and pre-motor activity (simple spikes) in individual Purkinje cells. This is further explained in the new section “The consequences of complex spike firing” in the Discussion.

Second, complex spikes are indeed immediately followed by a period of disrupted simple spike firing, although this period is typically only 10-20 ms in mouse Purkinje cells, with a few exceptions possibly lasting up to 200 ms (Zhou et al., eLife 2014, doi: 10.7554/eLife.02536). And, indeed, the presence of a complex spike during protraction is correlated with reduced simple spike firing & tongue extension (Fig. 6E-F). This is now also mentioned in the Discussion (section “The consequences of complex spike firing”).

Third, indeed, the signal propagation from Purkinje cells to neurons of the cerebellar nuclei is essential to understand the ultimate effects of Purkinje cell activity on the rest of the brain, and the behaviour of the animal. Nuclear activity during licking has been previously reported by Detlef Heck's lab (Lu et al., Front Neural Circuits 2013, doi: 10.3389/fncir.2013.00056). The activity patterns in the cerebellar nuclei during licking, however, were outside of the scope of our present study. Even so, when making the comparison with eyeblink conditioning, we can extrapolate that, different from increases in simple spike activity, synchronized complex spike activity can elicit a pause and subsequent rebound in the downstream nuclei neurons, and probably readily evoke well-timed conditioned reflexes (Ten Brinke et al., eLife 2017, doi: 10.7554/eLife.28132). We now make this comparison in the Discussion, highlighting indeed the circuit effect.

d) For the relevance to humans / clinical applications: Are tongue movements / sensations different in primates than in rodents (for example)? Same number of muscles? Taste receptors? Is there something to make the argument that tongue movement is particularly complex in humans, because of its role in language?

Authors' response:

We thank the Reviewer for this lovely question. We have added a new section to the Discussion, titled "Tongue movements in different species". In this section, we emphasize that the basic anatomy and musculature of (most) mammalian tongues are similar, but that there are species-specific adaptations to food sources, environment and behaviour. As a consequence, there are even differences in tongue anatomy between different mouse species, as well as between different primates. From this we take, as we now argue in the Discussion, that it is likely that the principles of cerebellar control of tongue movements are broadly similar between mammals, but that there are differences between species at a more detailed level.

Regarding the taste receptors: we feel that this is outside the scope of our study. We have nevertheless added a remark on the existence of differences in taste receptors between mammals in the section on tongue movements and inter-species differences in the Discussion, including references to a number of papers on this topic for interested readers.

Referee #2:

This manuscript by the De Zeeuw lab examines targeted tongue movements in the mouse and its relationship to Purkinje cell discharge. The basic observations are that both the simple spike discharge and complex spike discharge are modulated in relationship to tongue movements but with different modulation characteristics, as one would expect as Purkinje cells are modulated with virtually all movements. The study also uses optogenetic manipulations to test how activation of Purkinje cells modulates tongue movements. The work is interesting as the first detailed examination of this question. However, it's not quite clear what the overall purpose of the study was beyond an exploration of the topic and wish that there was a more specific hypothesis driving the work. Still the study has important value, as providing fundamental characteristics of Purkinje cell firing in relation to tongue movements.

Authors' response:

We thank Reviewer 2 for the constructive remarks and the overall positive evaluation of our manuscript. We have re-written the abstract to emphasize the research questions and findings, and to give a more structured expectation to future readers. We hope that also the new graphical abstract and accompanying legend will help to make the message of our study easier to understand.

1. There are several major questions and comments that need to be addressed by the authors. The computational model does not seem to add very much to the findings or understanding of the role of either complex spike or simple spike firing in tongue movements. The relationship revealed by the model seemed to be somewhat weak. Similarly, it is hard to understand the meaning of the decoding contribution of the Purkinje cell parameters. For example, the statement on line 337 says that 69% of Purkinje cells had at least some relation to licking behavior, with the word "some" not very precise. Also, given the modulation observed in the complex spikes, both at the beginning and end of the lick in Figure 2, it was surprising that the model found that complex spike firing was a poor predictor of licking. Also, what does it mean that in Figure 3A that there is no simple spike CV2 bias between licking and no licking? Or that there is no bias for simple spike frequency in Figure 2B. Also, the model was based on exceeding a SHARP threshold of 65. This needs to be explained better, including why that threshold was used, implications for statistical significance, and how to interpret.

Authors' response:

We are grateful for these critical remarks of Reviewer 2. They helped us to integrate the computational model better in the flow of our manuscript, and to phrase the modelling results more clearly. We have changed Figure 3 to better illustrate our results.

First of all, Purkinje cells in the region of the cerebellar cortex are involved in multiple tasks (see also comment a of Reviewer 1 and our answer to that). Hence, although we can (and did) establish that licking is associated with changes in Purkinje cell activity, we cannot infer the opposite – that changes in Purkinje cell activity are indicative of licking. After all, Purkinje cell activity can also reflect respiration, swallowing, limb movements, sensory processing, or even other processes. With our computational model we tested exactly this: how good can we predict whether a mouse is licking or not based on the activity pattern of a single Purkinje cell at any given moment. We have added a sentence to the Results (section "Decoding of Purkinje cell activity patterns with machine learning inference") to better motivate the inclusion of the computational model.

Second, after establishing that it is indeed possible to decode the Purkinje cell signal and predict the presence or absence of licking during any 200 ms interval, we asked which activity parameter had the strongest predictive value. As Reviewer 2 noted, we observed that complex spikes did not really allow us to differentiate between licking and non-licking, when considering 200 ms intervals. This is actually not so surprising, given the average complex spike rate of 1-2 Hz. This implies that most 200 ms intervals will encompass no complex spikes – licking or not, and, thus, the absence or presence of a complex spike during such a short time interval cannot be strongly predictive.

At first sight, this lack of impact of complex spikes may look as a flaw in our approach, but we think it does reflect the way complex spikes occur. Previously, we could demonstrate that complex spike rates are very hard to alter; the most one can achieve is a change in the timing of complex spikes (Negrello et al., PLoS Comp Biol 2019, doi: 10.1371/journal.pcbi.1006475; Ju et al., J Physiol 2019, doi: 10.1113/JP277413). Thus, the lack of impact of complex spikes in this particular analysis does not contradict our other findings on complex spike firing in relation to licking. We have clarified this in the Discussion.

Third, using the SHAP analysis, we can investigate the relative contributions of the three spiking parameters studied (complex spike frequency, simple spike frequency and simple spike CV2). As discussed above, the absolute SHAP values of complex spike frequency are low. The other two measures, simple spike frequency and simple spike CV2, show a heterogeneous pattern (see the revised Fig. 3D). CV2 is calculated by normalizing for the instantaneous firing, so that these two measures can (within boundaries) fluctuate independently from each other. To illustrate this, we have modified the panels 3A and 3B, and now demonstrate one Purkinje cell with a high value for simple spike frequency (Fig. 3A) and one with a high value for simple spike CV2 (Fig. 3B).

Fourth, Reviewer 2 remarked that the threshold of 65% was chosen arbitrarily and should be better motivated. To this end, we now include the distribution of prediction accuracies of bout and inter-bout intervals (Fig. 3C). From this new panel it is clear that the Purkinje cells form a continuum and that any threshold would be arbitrary. Hence, we show all Purkinje cells individually (Fig. 3D) and grouped in three categories (Fig. 3E). We think that the new presentation gives a deeper insight into the structure of the data.

2. Another concern is that the role of complex spikes in error detection or error signaling argued for in the Discussion, specifically as providing a sensory prediction error in the experiment in which the lick port was moved. For the experiments in which the click port was moved, the complex spike firing showed increases both to the right and the left but did not discriminate between the two directions. This lack of directionality suggests that the complex spikes are not error signalling as there is no information about the specifics of the movement but again possibly a change of state. Is this how the authors are interpreting this finding?

Authors' response:

We would like to thank Reviewer 2 for this remark. We agree that the initial description in the text, while factually correct, could be perceived as confusing. In the previous version, we used the correlation between the complex spike responses evoked by lick port movements to the right and to the centre. These are correlated, implying that Purkinje cells that respond

strongly to movement in one direction, tend to respond strongly to movement in the other direction as well.

This, however, does not imply that Purkinje cells cannot discriminate between rightward and leftward movements. To quantify and illustrate this, we have modified Fig. 7E. Focussing on the statistically significant complex spike responses ($Z > 3$; coloured shades in Fig. 7E), there is a general tendency to encode rightward movements stronger ($p < 0.0001$, Wilcoxon matched-pairs test). Also, when evaluating a laterality ratio, we observe that most Purkinje cells have a preference for either direction, rather than for both. Thus, there is directionality in complex spike encoding, and this was not represented well in the previous version. We have adapted the text accordingly. Moreover, we now also use this as an argument in the Discussion in favour of at least partly some level of error encoding by the complex spikes (next to the other forms of encoding that are also present; see below).

3. Instead, the findings suggest a role for the complex spikes in a change of state that was mentioned in the paper, which has also been suggested by Streng et al. Journal of Neuroscience, 2017 and Cerebellum in 2022. The change of state should be addressed given the present findings. Similarly, the fact that the complex spikes are show modulation related to the control of normal tongue movements without any obvious errors, also suggests that the complex spikes have a role in specifying movement parameters, as suggested by a number of recent papers. This issue should also be discussed in the manuscript.

Authors' response:

To address this comment, as well as comment b of Reviewer 1, we have added a section to the Discussion ("The consequences of complex spike firing"). In this section, we now (better) highlight the occurrence of complex spikes at behavioural state changes. We now cite both indicated references. In this section, we also address the relation between complex spike firing and movement parameters. Thanks to Reviewer 2 for the suggestions!

4. Concerning the topography of the responses shown in Figure 4B and D, asking the question whether there is a specific distribution of the simple spike and complex spike responses depends on that the sampling of Purkinje cells was uniform across the region of the cerebellar cortex studied. Reviewers need to know the locations of the Purkinje cells recorded on those maps to interpret those findings. Also, as only a small number of Purkinje cells were recorded in each mouse this may make it harder to make such a spatial determination and statistical testing is needed.

Authors' response:

This is a fair criticism. We have added the locations of the recordings to Fig. 4B and 4D.

As now explained in the Methods, we have intrapolated the signals of the electrodes to reduce the impact of unequal sampling. As suggested by Reviewer 2, we have now tested the lateral vs. the medial parts of Fig. 4B and 4D.

Specific Comments:

5. In the first Paragraph of the results, lines 279 to 283, the comment is made that Purkinje cells can be defined as task related when they display both complex spike and simple spike modulation. It was unclear why the manuscript makes this point or how it fits into the overall results and discussion. Especially as they only find 19% of the Purkinje cells having both complex spikes and simple spike modulation in relation to the tongue movements.

Authors' response:

This comment is similar to comment a of Reviewer 1, and we have addressed both comments in the new version of the Discussion. Please see our response to comment a of Reviewer 1 for more details.

6. Concerning that complex spike modulation in relation to tongue movements, there is an increase in complex spikes at the onset as shown in Figure 2A and B. The text then states in lines 309 and 310 that there's an increase in complex firing in 51% of Purkinje cells also at the end of the licking bout. However, in Figure 2C and D it appears the complex firing decreases, not increases. Please clarify. Also, for the plots in Figure 2B and D please label the time points that are actually significant modulation by the criteria used, as few of the points in D seem to ± 3 Zscores. Also, what exactly does the rectangular boxes in B&D signify? Also needed is when the licking starts and ends and the variability of those time points.

Authors' response:

We apologize for the confusion created by the initial version of Fig. 2, and we thank Reviewer 2 for pointing this out. The panels A-D all come from the same Purkinje cell. This Purkinje cell shows increased complex spike firing at the start, but not at the end of bouts. This behaviour is pretty common (see Venn diagram in Fig. 2E). We have now clarified in the figure legend that panels A-D originate from the same Purkinje cell and serve as an illustration. We have also altered the text of the Results to better account for the heterogeneity in Purkinje cell behaviour. The rectangles mark the time interval during which we tested for significance. This is now also clarified in the figure legend. We have added the thresholds for significance in panels B and D.

7. In Figure 4 and the associated analysis a coherence of greater than 0.5 was used as the threshold for a coherence. How was this threshold chosen and does it relate in any way to statistical significance? Also, in Figure 4F the claim is made that the simple and complex spikes are in antiphase. Was this based on a statistical test or is this a qualitative statement?

Authors' response:

We thank the Reviewer for this important question. The main purpose of Fig. 4 is to illustrate the distribution of coherence values between complex spike firing, simple spike firing and licking. The threshold of 0.5 is admittedly chosen arbitrarily. We use this threshold to facilitate the description in the text, but illustrate (also) the non-thresholded values and we use these for our statistical analysis. We have now toned down the implication of this threshold in the text and figure legend, and relate more to the actual distribution.

The observation that more Purkinje cells show antiphase relations between complex spike and simple spike firing is based on Fig. 4F and was initially a qualitative observation. The cautious phrasing in the text of the Results ("In general, though, the majority of Purkinje cells had a tendency to fire complex spikes and simple spikes in antiphase.") is in line with the gradual distribution of the Purkinje cells in Fig. 4F. Of the 57 Purkinje cells with a

complex spike-simple spike coherence >0.5, 35 (61%) are in the lower half of the graph of Fig. 4F.

8. In Figure 8 the optogenetic stimulation shows modulation in both the tongue movement and the simple and complex firing. Did the optogenetic stimulation generate any other movements or twitches, particularly in the oral facial areas, given that the Purkinje cells were targeted widely. Stimulation 10-90 ms (I) is in-between licks, while 90-170 ms (J) is during lick execution and the effects are similar. This sort of confounding when it comes to control mechanisms, can the authors comment? Also the 10-90 ms stimulation reduces the next lick probability. Is this consistent with CS involvement in lick bouts?

Authors' response:

Optogenetic stimulation can indeed evoke muscle twitches. We have dedicated a new section in the Discussion ("Multitasking in the cerebellum") to cerebellar involvement in different types of behaviour. For the rest, we focussed on the modulation of ongoing movement rather than on twitches for this study. We would like to emphasize that tongue movements during rhythmic licking are continuous. The 10-90 ms interval is, therefore, typically during tongue retraction, not in between licking. Our results demonstrate that optogenetic Purkinje cell stimulation during either tongue retraction or tongue protraction leads to an ipsilateral bias of the consecutive lick. This is corroborated by the impact of the 10-170 ms stimulation, thus spanning both retraction and protraction. It would indeed be interesting to study the impact of cerebellar activity on the coordination of individual tongue muscles in future studies. It should be noted, though, that 10-90 ms and 90-170 ms stimulation are not identical. The reduction in tongue extension (y-axis in Fig. 8H-K) occurs more during lick 1 for 10-90 ms and during lick 2 for 90-170 ms. This may be explained by the occurrence of increased complex spike firing around 100 ms after the start of optogenetic stimulation (in line with Fig. 6F).

Dear Dr Bosman,

Re: JP-RP-2024-287732R1 "Cerebellar control of targeted tongue movements" by Lorenzo Bina, Camilla Ciapponi, Si-yang Yu, Xiang Wang, Laurens W. Bosman, and Chris De Zeeuw

We are pleased to tell you that your paper has been accepted for publication in The Journal of Physiology.

Yours sincerely,

Harold Schultz
Senior Editor
The Journal of Physiology

If you would like to receive our 'Research Roundup', a monthly newsletter highlighting the cutting-edge research published in The Physiological Society's family of journals (The Journal of Physiology, Experimental Physiology, Physiological Reports, The Journal of Nutritional Physiology and The Journal of Precision Medicine: Health and Disease), please click this link, fill in your name and email address and select 'Research Roundup':
<https://www.physoc.org/journals-and-media/membernews>

- **TRANSPARENT PEER REVIEW POLICY:** To improve the transparency of its peer review process, The Journal of Physiology publishes online as supporting information the peer review history of all articles accepted for publication. Readers will have access to decision letters, including Editors' comments and referee reports, for each version of the manuscript as well as any author responses to peer review comments. Referees can decide whether or not they wish to be named on the peer review history document.
- You can help your research get the attention it deserves! Check out Wiley's free Promotion Guide for best-practice recommendations for promoting your work at: www.wileyauthors.com/eeo/guide. You can learn more about Wiley Editing Services which offers professional video, design, and writing services to create shareable video abstracts, infographics, conference posters, lay summaries, and research news stories for your research at: www.wileyauthors.com/eeo/promotion.
- **IMPORTANT NOTICE ABOUT OPEN ACCESS:** To assist authors whose funding agencies mandate public access to published research findings sooner than 12 months after publication, The Journal of Physiology allows authors to pay an Open Access (OA) fee to have their papers made freely available immediately on publication.

Reviewing Editor's comments:

The authors have addressed all the reviewers' concerns and made a great improvement. I have no further questions.

Senior Editor's comments:

The editors wish to thank the authors for these revised adjustments to the manuscript. The article is now accepted for publication. Congratulations for an interesting and insightful study. Please consider the Journal of Physiology for your future studies.

Referee #1:

All my previous concerns have been appropriately addressed.

Referee #2:

The authors have made a very through effort in addressing the concerns and issues with the the original draft. The work is the first to investigate the cerebellum's role in tongue movement using electrophysiological/behavioral perspective. The work is therefore original, well done and the conclusions appropriate. I was particularly impressed by the discussion with its review of the tongue.

END OF COMMENTS